# The role of operators in sustainable whale-watching tourism: Proposing a continuous training framework

**Alice Affatati**[1,2], **Chiara Scaini**[1], **Anna Scaini**[3,4]*

**1** National Institute of Oceanography and Applied Geophysics - OGS, Trieste, Italy, **2** Department of Mathematics, Informatics and Geosciences, University of Trieste, Trieste, Italy, **3** Department of Physical Geography, Stockholm University, Stockholm, Sweden, **4** Bolin Centre for Climate Research, Stockholm University, Stockholm, Sweden

* anna.scaini@natgeo.su.se

## Abstract

Whale watching is considered a form of green tourism, but can affect marine ecosystems, impacting cetaceans' behavior and potentially increasing acoustic pollution. A more sustainable whale-watching practice should employ a comprehensive approach involving all stakeholders, but whale-watching operators are rarely involved. We propose a method to assess whale–watching operators' perceptions regarding the possible effects of their activity on marine fauna and preferred mitigation solutions, by means of online questionnaires and website communication strategies. Results from Canadian whale-watching operators show that they observe regulations regarding distance to whales but only partially perceive general vessels' impacts on fauna. Three recognized whale-watching experts identify the need for continuous training targeted at operators, which should include the impacts on marine ecosystems. A continuous training framework is proposed that targets whale-watching operators in addition to tourists, and involves scientists in several steps of the approach. This study serves as a starting point to involve operators' in order to advance towards a sustainable whale-watching tourism.

## 1. Introduction

Tourist activities impact ecosystems directly, e.g., through habitat degradation, pollution, and loss of biodiversity, and indirectly by affecting ecosystem services provision, and despite the rise of ecotourism practices [1–4]. Enhancing ecotourism is crucial to mitigate impacts on ecosystems, particularly on marine habitats, since they constitute a natural capital crucial for the health of human and marine ecosystems [5, 6].

Among ecotourism activities, whale watching has been increasing worldwide since the early ´90s, generating a wide range of opportunities and economic benefits for coastal communities [7, 8]. Hoyt [9 p3] defined whale watching, as "tours by boat, air or from land, formal or informal, with at least some commercial aspect, to see, swim with, and/or listen to any of the 83 species of whales, dolphins and porpoises". Whale watching, as an ecotourism activity,

**Data Availability Statement:** All relevant data are within the paper and its Supporting Information files.

**Funding:** AA acknowledges support from the National Institute of Oceanography and Applied Geophysics - OGS, the University of Trieste, JASCO Applied Science. AS acknowledges support from the Swedish Research Council for Sustainable Development Formas (grant 2022-00329). The funders had no role in study design, data collection and analysis, decision to publish, or preparation of the manuscript.

**Competing interests:** The authors have declared that no competing interests exist.

should be a non-consumptive, educational, sustainable experience providing an important conservation message that excludes exploiting cetaceans [10–12]. However, the increasing number of tourists [7] has led whale-watching excursions to potentially affect marine habitats and fauna in multiple ways [13, 14]. In particular, motorized vessels, usually used for whale watching, contribute to acoustic pollution and can impact marine mammals causing behavioral responses such as avoidance or attraction to boats and changes in surfacing patterns [15–19]. In addition, a growing debate that these tourism activities could be harmful to whales has created disagreement among researchers and highlighted the need for more assessments of these impacts [20–24].

The sustainability of whale-watching tourism is currently limited by several factors, including poor implementation of scientific recommendations and ineffective communication about the negative impacts of these activities on fauna [25]. There is a wide consensus that a lack of best practices implementation and improper management can impact cetaceans and disrupt tourist experiences [26–29]. Olszewski-Strzyzowski [24] identified good practices for promoting sustainable tourism, improving communication with tourists, and minimizing environmental impacts. In order to reduce impacts and promote innovation in the whale-watching field, researchers and operators should co-design activities and share knowledge [26].

Communication impacts tourist expectations about the activity. Websites are increasingly used by whale-watching companies as tools to reach a broad, international audience informing tourists and tailoring expectations before the trip [30–34]. Multiple researchers have sought to analyze content from websites for promoting nature-related tourism [31–33], but there are no defined standards or globally-accepted methods to evaluate website content [35–37]. A tool to evaluate web communication in nautical tourism (defined as one of the components of marine tourism) was proposed to include completeness, correctness, and accuracy of website content [35]. There is a need for more studies analyzing if aspects related to sustainable tourism are effectively communicated on whale-watching websites [e.g., 33].

In order to develop sustainable whale-watching activities, it is paramount to understand the perception of whale-watching impacts on ecosystems and involve a broad range of stakeholders in order to improve the code of conduct and enhance mitigation strategies [26, 38, 40]. Stakeholder participation should be broadened to include tourists, scientists, industries, regulatory bodies, non-governmental organizations, and whale-watching operators [39, 40]. Among the stakeholders involved in whale-watching activities, operators play a critical role in the practical activity, and communicating the importance of marine fauna conservation and guiding and inspiring tourists (e.g., [40, 41]). Most research on whale-watching tourism focuses on questionnaires delivered to tourists [42–45] with close to no research studies on whale watching developed specifically for operators. An exception is the work of Tepsich and colleagues [42], who focused on surveying different whale watching categories around the Pelagos Sanctuary in Italy, categorizing them as "commercial whale-watching", "cetacean ecotourism", or "research whale watching" (after [42]) depending on trip length and research components. Whale-watching operators' satisfaction in conducting the activity is recognized as critical to achieving sustainability [42, 46–48].

Here, we address the point of view of whale-watching tourist companies regarding interactions between vessels and marine fauna and possible solutions to mitigate impacts. This study investigates and evaluates whale-watching operators´ perception of the impacts on marine fauna related to their activities, and proposes a framework for continuous training devoted to whale-watching operators and involving scientific knowledge. A novel compound approach consisting of a combination of questionnaires and data obtained from website analysis is used to tackle the following aspects:

1. what whale-watching operators know about the acoustic impact of their activity on marine mammals,

2. how whale-watching operators communicate through their websites regarding tourism activity and its impacts on the marine ecosystem,

3. strategies to enhance the sustainability of whale-watching tourism with selected experts in the field.

The analysis focuses on Canada at a national scale and selected provinces characterized by different degrees of tourism development and the presence of different marine species, including endangered ones. Since the 1990s, the presence of 30 whale species has led to the growth of the whale-watching industry in Canada [7]. However, whale populations are depleted due to anthropogenic stressors, e.g., underwater noise and pollution [49], and the Canadian government has thus strengthened its regulations to protect cetaceans and increase tour operators' awareness of potential impacts [50]. The low number of questionnaire responses from whale-watching operators prevented us from developing statistical analyses and a more comprehensive scrutiny of the results, yet this blend of methods represents a new compound approach that can be extended to other areas.

## 2. Materials and methods

Through online surveys and the analysis of whale-watching companies' websites, we gathered information about whale-watching operators' perspectives regarding vessels' impacts and their communication strategy (described in **sections 2.1** and **2.2**, respectively). The opinion of whale-watching experts was also collected using a separate questionnaire (**section 2.3**).

### 2.1 Whale-watching operators' questionnaire

A questionnaire was designed to investigate whale-watching operators' perception of impacts from vessels and its effect on marine mammals. A Google search (using the string "Whale-watching companies Canada") conducted during January and February 2022 allowed us to identify ninety-one Canadian whale-watching companies. Operators working in these companies were invited via email to fill in the online questionnaire (the list of questions is shown in S1 Table). The questionnaire was prepared using Google Forms (available at S1 Text) and was distributed on February 1, 2022; one reminder was sent on March 31, 2022. The questionnaire consisted of twenty multiple-choice or checkbox questions - all compulsory - associated with unique identifiers (ID) and subdivided into three sections (**Table 1**):

- **Introductory questions**: designed to gain insight into the main characteristics of whale-watching tours, including selection of areas where the company operates, see **Fig 1.** Introductory questions are not included in **Table 1**;

- **Trip-specific questions** (ID 1,2): used to collect data on the whale-watching trips, in particular the distance kept from cetaceans, the interaction between vessels and marine fauna, and the tourists' preferences during trips;

- **Perception of impacts questions** (ID 3–5): ID 3 was prepared to evaluate operators' perception and knowledge of vessels' general impacts. Vessels impose a variety of impacts on marine ecosystems. Chemical pollution from vessels is produced mainly by fuel, oil, and anti-fouling treatments discharged from motorized vessels and can reduce water quality or enhance accumulation in sediments. While 4-stroke engines are cleaner, 2-stroke engines, usually used on small boats, produce a higher amount of chemical pollution [51]. Moreover,

**Table 1. Summary of the topics, questions, and options given in the whale-watching operators' questionnaire.**
Each question is associated with a unique identifier (ID). ID 3–5 were multiple answer questions.

| TRIP-SPECIFIC QUESTIONS | | QUESTION ID |
|---|---|---|
| "How far should the vessel be from the cetacean, at least?" | • 100 m<br>• 200 m<br>• 300 m<br>• 350 m<br>• I don't know<br>• It doesn't matter | ID 1 |
| "How far does your vessel go from the whale during the tours?" | • More than 100 m<br>• More than 200 m<br>• Less than 10 m<br>• 10–50 m<br>• Around 100 m<br>• I don't know | ID 2 |
| PERCEPTION OF IMPACTS QUESTIONS | | |
| "Do you think that there are any issues related to the interaction between ships and marine fauna? Select all that might apply." | • Chemical pollution<br>• Ship strikes<br>• Noise<br>• Introduction of invasive species<br>• Biofouling<br>• I don't know<br>• None of them<br>• All of them | ID 3 |
| "What are the most important aspects for tourists? Select all that might apply." | • Going very close to the whales<br>• Being taught something about the biology/ecology of the whales<br>• Seeing as many animals as possible<br>• Getting to know something about the marine environment of the area<br>• Seeing at least one whale during the trip<br>• Other<br>• I don't know | ID 4 |
| "Which of these strategies would you prefer to follow in order to reduce the impact of underwater noise from shipping on cetaceans? (Select all that might apply)" | • Implement mandatory AIS on all vessels (even small recreational ones)<br>• Increase distance with whales<br>• Decrease the speed of the vessel near whale migratory routes or selected areas<br>• Increase the number of marine protected areas with limited access for vessels with a permit<br>• Implement mandatory avoidance of feeding and breeding areas during the most important times of day<br>• Diminish the duration of the whale-watching trips<br>• Avoid whale-watching tours during specific times of day<br>• Increase the duration of the whale-watching trips, but reduce the trips to one, daily | ID 5 |

alien species might be transported on hulls creating a major conservation issue [52]. Ship-strikes can cause whale mortality and have been increasing worldwide due to the increase in ship speed [53]. In addition, shipping noise is of great concern due to a steady increase in traffic. Underwater noise produced by motorized vessels can hinder marine fauna behavior and induce acoustic masking [54]. In order to tackle this issue and find efficient mitigation measures, all the stakeholders involved should be at least aware of the problem. ID 4 and 5 strictly referred to whale-watching activities.

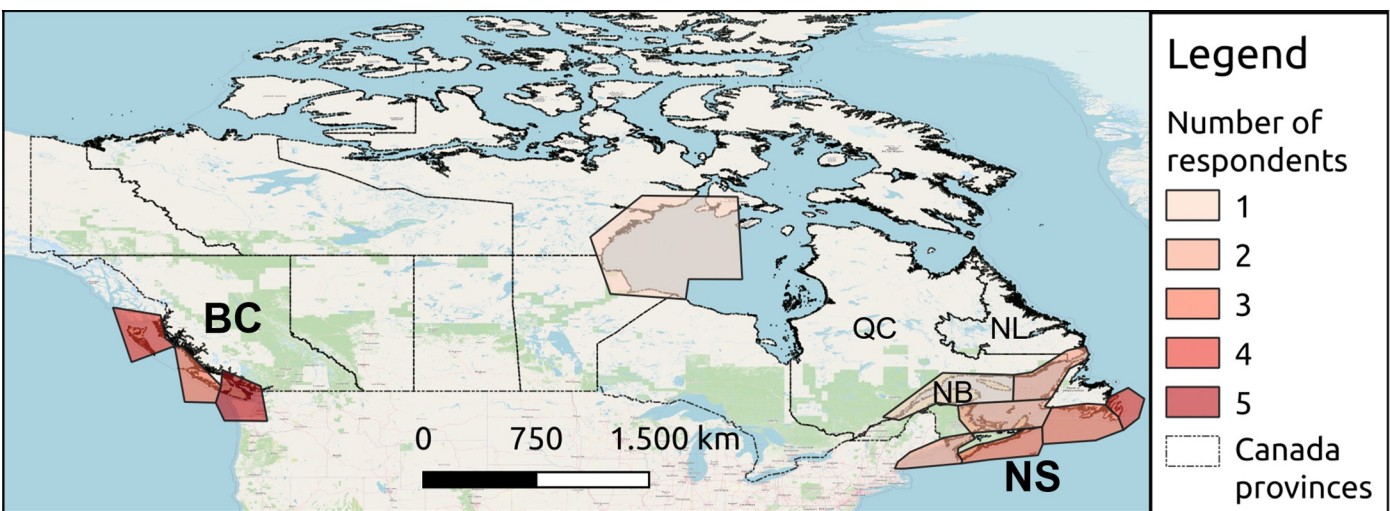

**Fig 1. Map showing the number of operators who have responded to the whale-watching operators´ questionnaire, assigned to each area where they operate according to their response in the introductory questions.** The polygons of the areas where companies operate were defined by the authors, georeferenced manually and plotted using QGIS Open source software (https://docs.qgis.org/)). The provinces of British Columbia (BC) and Nova Scotia (NS), where most responses were collected, are highlighted in bold. The Canada provinces were extracted from the Database of Global Administrative Areas (GADM, www.gadm.org), version 2.5, July 2015 which provides them freely for academic and non-commercial use. The results of the website analysis include also the provinces of Québec (QC), New Brunswick (NB) and Newfoundland and Labrador (NL) (Table 5). *Map data from OpenStreetMap available from https://www.openstreetmap.org (Openstreetmap contributors, 2023) under the Open Data Commons Open Database License (ODbL)* [56].

In order to avoid influencing the participants, we did not provide any explanation or information on the topic of underwater noise [54]. In designing the questionnaire, we have addressed social desirability bias, e.g., the tendency of respondents to bias their choices to comply with social norms. For instance, in questions ID1 and ID2 we added options with values similar, but not equal, to the correct ones [55]. In question ID1 We selected 100 and 200 m based on current guidelines and legislations. Canada's Marine Mammal Regulations mention "keeping a minimum of 100 meters away from most whales, dolphins, and porpoises, and keeping a minimum of 200 meters away if they are in resting position or with their calf". Other reasonable distances in the upper and lower bound of the prescribed distance (displaying them in the question in an increasing order) were included in order to assess operators' knowledge, and to partially prevent biased responses. In ID2, we did not choose an increasing scale to try to avoid bias and discourage respondents towards selecting an intermediate number. The consent form was designed to highlight subject anonymity, which is another common measure to reduce biased responses.

**2.1.1 Ethics statement.** The collected data was analyzed in aggregated form, and all participants gave consent before completing the questionnaire. The authors ensure compliance of the proposed research with EU and Canadian legislation on ethics in research. The research protocol and questionnaires were prepared based on input from the Ethics support function at Stockholm University, in order to ensure no sensitive data would be processed. The research involved exclusively non-identifiable human participants who participated in an online questionnaire. All participants were volunteers, over 18 years old, and gave written consent to participate by accepting the conditions included in the questionnaire form. The research did not involve any sensitive personal data, as defined in the EU General Data Protection Regulation (GDPR). Therefore, ethical review and approval was not required for the study as it relied exclusively on publicly available information or secondary use of anonymous information.

The study also benefitted from the contribution of selected whale-watching experts who filled a specific form after giving written consent to the form conditions and agreed to be cited in the paper.

In addition:

- the information being processed does not reveal racial or ethnic origin, political opinions, religious or philosophical beliefs, or trade union membership;

- the information being processed does not contain genetic data, biometric data for the purpose of uniquely identifying a natural person, data concerning health, or data concerning a natural person's sex life or sexual orientation;

- the information being processed does not regard violations of law that include crimes, judgments in criminal cases, penal law sanctions, or administrative deprivations of liberty.

## 2.2 Analysis of whale-watching companies' websites

The websites of the 91 whale-watching companies identified in section 2.1 were manually analyzed. We focused on the information that should be included in high-quality tourism websites in order to make it complete, correct, and accurate [36]. Based on [36], we defined a set of specific information related to whale-watching activities deemed positive in terms of content quality. The underlying hypothesis is that each of the identified factors positively contributes to a better communication of the whale-watching activities to the public. Information ranges from the type of scientific-based information provided (e.g., marine biology notions) to best practices and potential impacts of misbehaviors. This includes both text and images that can support or demonstrate concepts (e.g., images of encounters with cetaceans). The presence of the following information was assessed:

- Is this whale-watching company a member of an environmental organization? (1 = Yes, 0 = No)

- Is there a mention of impacts on marine mammals? (1 = Yes, 0 = No)

- Is there a mention of best practices implemented to reduce impacts on marine mammals? (1 = Yes, 0 = No)

- Are there any links or information on marine biology or ecology of marine fauna that can be encountered in the area during the tours? (1 = Yes, 0 = No)

- Does the company declare to respect the required distance to be kept with cetaceans (note that it should be at least 100 m according to Canadian Marine Mammal Regulations)? This information was inferred based on website pictures showing clear closeness to a cetacean or website statements (e.g., mentioning a close distance or code of conduct). (1 = distance kept; 0 = distance not kept; n.a. = no mention)

- Is there a mention of a code of conduct implemented to protect marine fauna? (1 = Yes, 0 = No)

Following the aforementioned list, all websites sections were navigated manually to identify the information, ranked with 0 or 1 depending on its presence/absence. We defined a Sustainable Communication Index (SCI) calculated as the sum of the rating in each area. The SCI was calculated for all websites and those in each considered Canadian province (**Fig 1**). The normalized SCI was computed by dividing the SCI by the number of websites inspected. The

operation was done for each area and for all websites. This allows comparing the scores obtained for the different areas considered.

## 2.3 Whale-watching experts' questionnaire

An online Google Forms questionnaire was prepared to collect the opinions of three whale-watching experts on tourists' expectations and the importance of increasing and improving education and outreach for whale-watching operators (**Table 2**). More personal, semi-structured interviews with the three experts could provide a higher level of insights and reflection, which was only partly achieved using free-text options. The questionnaire was shared with Dr. Eric Hoyt (Research Fellow, Whale and Dolphin Conservation), Dr. Heidi Pearson (Associate Professor of Marine Biology, University of Alaska Southeast), and Ted Cheeseman (expedition leader, Cheesemans' Ecology Safaris; Co-Founder & Director, Happywhale). Dr. Hoyt and Dr. Pearson authored numerous papers and book chapters on whale-watching activities [9, 44]. Cheesemans' Ecology Safaris is a company specializing in responsible tourism (https://cheesemans.com/about). The participants agreed to participate in the questionnaire and be cited in the paper (see **Ethics statement, section 2.1.1**).

The questionnaire was organized into two main topics:

- **Interactions between ships and marine fauna:** we asked the experts to select issues related to the interaction between ships and marine fauna.

- **Experts' opinion on tourists' priorities:** we asked the experts to identify the most important aspects for tourists during whale-watching trips.

First, we asked the same questions as in the whale-watching operators' questionnaire, e.g., question ID 3 and question ID 4 (**section 2.1**). Then, we asked them to comment on the results of the corresponding question in the whale-watching operators' questionnaire (**section 2.2**).

The complete list of questions and options is shown in **S2 Table,** while selected questions are shown in **Table 2**.

**Table 2. Whale-watching experts' questionnaire.** Each question from the whale watching operators' questionnaire (Table 1) is referred to the corresponding identifier in the whale-watching operators' questionnaire (ID). Questions whose results were shown to experts are indicated with an asterisk. The full questionnaire is available in S2 Table and S2 Text.

| INTERACTION BETWEEN SHIPS AND MARINE FAUNA | | QUESTION ID |
|---|---|---|
| Only 4% selected 'All of them'. The least selected options were biofouling and chemical pollution. Why do you think some of the options were not selected? Select all that might apply. (*) | • Low awareness of the impact<br>• Limited knowledge of the subject<br>• Both of the above<br>• I don't know | ID 3 |
| **YOUR OPINION ON TOURISTS' PRIORITIES** | | |
| What are the most important aspects for tourists during whale-watching trips? Select all that might apply. | • Going very close to the whales<br>• Being taught something about the biology/ecology of the whales<br>• Seeing as many animals as possible<br>• Getting to know something about the marine environment of the area<br>• Seeing at least one whale during the trip<br>• I don't know | ID 4 |

## 3. Results

### 3.1 Analysis of whale-watching operators' questionnaire

The questionnaire was sent to the 91 whale-watching companies identified in **section 2.1**. Twenty-six operators working in these companies completed it. The number of respondents operating in each coastal area of Canada is shown in **Fig 1**.

In the following, we present the results for operators in the Canadian provinces of British Columbia (BC) and Nova Scotia (NS), highlighted in red in **Fig 1**. We analyze results specifically for BC and NS because they had the highest response rates in the questionnaire, 12 for BC and 5 NS, respectively (the number of respondents in the remaining provinces was 9). They are situated on the Pacific and the Atlantic Ocean, respectively, and could help represent the perception of whale watching potential impacts on each coast. Both provinces host endangered cetaceans: e.g., Southern Resident Killer Whales in BC and the North Atlantic Right Whales in NS.

In general, we found good knowledge and observance of regulations, with most operators observing the minimum distance of 100 m. All participants operating in BC and NS answered that their vessel keeps a distance of around 100 m, or more, from the whales during tours (**Table 3**).

**Table 4A** shows the perceived issues related to vessels' interaction with marine fauna. Only 1 operator in BC and 1 in NS chose the correct option, "All of them". Biofouling (n = 2) was the least perceived issue in both BC and NS. Most of the other issues had a similar percentage of responses in BC and NS, except for chemical pollution, which was indicated 8 times in BC and 1 time for NS. From the answers collected, the perception of vessel noise and ship strikes is high and is perceived similarly in BC and NS.

**Table 4B** shows tourists' preferences according to whale-watching operators. The most common answers to this question for BC were "Being taught something about the biology/ecology of the whales" (n = 12) and "Getting to know something about the marine environment" (n = 12). "Seeing at least one whale" was chosen by 10 operators. NS operators selected the same two options, with n = 5, n = 6, and n = 4, respectively.

Regarding the strategies to lessen underwater noise impacts, BC operators mainly chose the options to decrease the speed of the vessel (n = 9) and increase the number of marine protected areas (n = 4), **Table 4C**. In NS waters, operators indicated in their responses that they decrease the vessel's speed (n = 5) and implement mandatory avoidance of feeding and breeding areas during the most important times of day for these activities (n = 2).

### 3.2 Analysis of whale-watching websites

The results from the website analysis including the scores of each indicator and the SCI are shown in **Table 5**. The provinces with higher SCI values were BC and Québec (QC). Provinces

**Table 3. Results to questions ID 1 and 2 for BC and NS participants.**

| ID | Question | BC [n = 12] | NS [n = 5] |
|---|---|---|---|
| ID 1 | "How far does your vessel go from the whale during the tours?" | • "More than 200 m":6 <br> • "More than 100 m":5 <br> • "Around 100 m": 1 | • "More than 100 m": 1 <br> • "Around 100 m": 4 |
| ID 2 | "How far away should the vessel be from the cetacean at least?" | • "At least 100 m": 11 <br> • "At least 200 m": 1 | • "100 m": 5 |

**Table 4.** Responses of operators based in BC (left column), NS (middle column) and the total in Canada (right column) related to interactions between vessels on marine fauna (A), tourists´ preferences (B), and preferred solutions for mitigation (C). MPAs refers to Marine Protected Areas.

| A | | | | |
|---|---|---|---|---|
| **Opinions on general impacts from vessels** | **Options** | **BC (n = 37)** | **NS (n = 13)** | **Total (n = 50)** |
| | All of them | 1 | 1 | 2 |
| | Ship strikes | 11 | 4 | 5 |
| | Noise | 11 | 4 | 15 |
| | Introduction of invasive species | 4 | 2 | 6 |
| | Chemical pollution | 8 | 1 | 9 |
| | Biofouling | 2 | 1 | 3 |
| **B** | | | | |
| **Tourists preferences according to whale-watching operators** | **Options** | **BC (n = 40)** | **NS (n = 18)** | **Total (n = 58)** |
| | Being taught something about the biology/ecology of the whales | 12 | 5 | 17 |
| | Getting to know something about the marine environment of the area | 12 | 6 | 18 |
| | Going very close to the whales | 2 | 1 | 3 |
| | Seeing as many animals as possible | 4 | 2 | 6 |
| | Seeing at least one whale during the trip | 10 | 4 | 14 |
| **C** | | | | |
| **Preferred solutions for mitigation** | **Options** | **BC (n = 24)** | **NS (n = 11)** | **Total (n = 35)** |
| | Increase the number of MPAs with limited access for vessels with a permit | 4 | 1 | 5 |
| | Decrease the speed of the vessels near whale migratory routes or selected | 9 | 5 | 14 |
| | Increase distance from the whales | 2 | 0 | 2 |
| | Implement mandatory avoidance of feeding and breeding areas during the most important times of day for these activities | 2 | 2 | 4 |
| | Implement mandatory AIS on all vessels (even small recreational ones) | 5 | 3 | 8 |
| | Diminish the duration of the whale-watching trips | 1 | 0 | 1 |
| | Avoid whale-watching tours during specific times of day | 1 | 0 | 1 |

with less than five websites, e.g., Manitoba (MB), Prince Edward Island (PE), and Nunavut (NU), were not included.

Among all Canadian provinces, websites advertising whale-watching trips in BC displayed the highest amount of information consistent with the regulations, e.g., had the highest

**Table 5. Results of the website analysis including the scores of each indicator for the provinces of British Columbia (BC), Québec (QC), New Brunswick (NB), Nova Scotia (NS) and Newfoundland and Labrador (NL).** Rows show the website indicators investigated, and the normalized Sustainable Communication Index (SCI); columns show the provinces.

| Indicator | BC | QC | NB | NS | NL | Total |
|---|---|---|---|---|---|---|
| **Members of environmental organizations** | 19 | 2 | 0 | 3 | 2 | 26 |
| **Mention of impacts on marine mammals** | 9 | 1 | 1 | 0 | 4 | 15 |
| **Best practices to reduce impacts on marine mammals** | 11 | 3 | 1 | 2 | 1 | 18 |
| **Information on biology or ecology of fauna** | 22 | 3 | 1 | 9 | 4 | 39 |
| **Mention of distance kept to whales (>100m)** | 11 | 0 | 0 | 1 | 1 | 13 |
| **Mention of code of conduct** | 17 | 2 | 3 | 2 | 1 | 25 |
| **Websites inspected** | 36 | 6 | 5 | 19 | 16 | 82 |
| **Sustainable Communication Index (SCI)** | 89 | 11 | 6 | 17 | 13 | 136 |
| **SCI (normalized)** | 2.5 | 1.8 | 1.2 | 0.2 | 0.8 | 6.5 |

normalized SCI. In BC, 75% of websites advertising whale-watching trips did not mention potential impacts of vessels on marine mammals, whilst in NS no websites mentioned impacts on fauna. Only 31% of websites mentioned best practices for BC, and 11% for NS (**S1A Fig**). In BC, 33% of websites mentioned keeping a distance of more than 100 m from marine mammals, while 14% showed evidence of close proximity to the whales and 53% did not mention the issue (S1B Fig). As for NS, in 5% of the websites we see that the distance kept is consistent with regulations (> 100 m), while 42% showed evidence of close proximity to the animals (S1C Fig).

### 3.3 Whale-watching experts' opinions

**3.3.1 Interactions between ships and marine fauna.** Results from the operators' questionnaire indicated that only 4% selected "All of them", while the least selected options were biofouling and chemical pollution. Three whale-watching experts were asked to comment on these results. Eric Hoyt stated that "There is much more evidence for ship strike and noise as issues affecting marine fauna, but this depends on where your operator is located". The other two experts stated that there is limited knowledge and low awareness of the impacts of vessels on marine fauna. The three experts suggested that education and outreach activities directed to operators would increase their awareness of the potential impacts of whale-watching vessels and motivate them to reduce them. Ted Cheeseman added that education and outreach could be potent when delivered engagingly. Eric Hoyt maintained that operators and other stakeholders should devise guidelines cooperatively, connecting education and hands-on outreach.

**3.3.2 Expert opinion on tourists' priorities.** The experts chose what they believed to be the most critical aspects for tourists during whale-watching trips among the provided options, and added additional aspects including "enjoying the camaraderie of watching wildlife in a good group", "getting out on the water", and "feeling like they experienced something special" (**S3 Table**). The experts responded that education and outreach regarding tourism impacts on marine fauna would help operators manage tourists' expectations, e.g., in case they do not encounter a whale.

## 4. Discussion

Our work combined sources of information through perception data from questionnaires to whale-watching operators, website content and experts' opinions. The responses collected from whale-watching operators are not statistically representative of the entire population of whale-watching operators in Canada, preventing us from conducting a statistical analysis of the results, but nonetheless provide useful insights to understand their point of view (section 4.1). A Sustainable Communication Index is introduced as a novel tool to analyze the online whale-watching communication strategy (section 4.2). The questionnaire for whale-watching operators can be adapted with minimum changes to different settings and case studies, creating a pliant method that, together with the websites' analysis, can complement the existing works on tourists' perspectives of whale-watching activities. Further work including interviews and direct interaction with whale-watching operators is pivotal to improve operators' perceptions of overall impacts on marine mammals and move towards a continuous training framework (presented in section 4.3) supporting the development of a more sustainable tourism activity.

### 4.1 Operators' perception of tourists' priorities, whale-watching impacts, and mitigation strategies

Online questionnaires were chosen because of their time and cost efficiency, flexibility, lowered interviewer bias, and due to the unrestricted geographic coverage [57]. In addition to the

low number of responses, some limitations exist regarding respondent truthfulness [57]). Moreover, results might be subjected to a degree of bias, in particular due to social desirability [55]. The responses collected from the questionnaires indicate that whale-watching operators perceive tourists as interested in learning about the environment they explore during the tours (**sections 3.1** and **3.2**). In contrast, the options "availability of knowledgeable guides/staff" and "interesting information about wildlife" did not score highly among the provided options in a questionnaire tourists engaged in wildlife tourism activities in Australia and New Zealand [58]. However, the study on improving wildlife tourism sustainability highlighted "a substantial correlation between the amount visitors believed they learned about the wildlife during their visit and their overall satisfaction with the wildlife experience" [58 p8].

Only a few whale-watching operators believe that the most crucial aspect for tourists is seeing a whale during the trip (**section 3.1**). In a study conducted in Moreton Island, Australia, 35% of whale-watching tourists were satisfied with the trip even without encountering whales [45]. For tourists, proximity to the animals was not the only factor determining an enjoyable trip; seeing a large number of whales and admiring them performing spectacular behavior were the two most important options. However, responding to a similar questionnaire, tourists indicated getting close to whales as a decisive factor to determine trip quality [44]. Some studies (e.g., [59]) identify two types of tourists: specialized whale watchers, who gave the lowest importance to observing whales, and recreationist whale watchers who gave the lowest rate of importance to "whale culture and preservation". Given the complexity of the relation between tourists and whale observations, further research should assess the factors contributing to a satisfactory tourist experience.

In BC and NS, most operators are aware of whale-watching impacts of vessel noise and ship strikes, but other aspects (e.g., biofouling, chemical pollution) are less perceived (**Table 4A**). Impacts produced by biofouling and chemical pollution are perceived similarly by companies operating in BC and NS (**Table 4A**), showing an overall low awareness of general impacts of vessels on marine fauna-likely due to lack of information-, but an overall high awareness of impacts caused by vessel noise and ship strikes. Since vessels impact marine ecosystems [51–54], a more comprehensive view of the general impacts might help operators moving towards a more sustainable practice without hindering the economic side of the activity. Surveyed experts say that the increase in perception is either due to more evidence of ship strikes and noise in some locations (Eric Hoyt), or due to low awareness and knowledge of the topic (Heidi Pearson and Ted Cheeseman). Another factor that might influence perception is that Canada's public outreach activities are mostly devoted to acoustic pollution, with less attention to other impacts [60].

More than half of the analyzed Canadian whale-watching websites mention existing rules regarding keeping a minimum or suggested distance from the cetaceans, which suggests that they are prone to observe them (**Table 5**). BC operators claim to be more conservative when approaching animals at a distance greater than 100 m compared to NS operators. The Fisheries and Oceans Canada (DFO) regulations are different in case endangered species are present in the area [61]. Vessels should keep a minimum of 100 m from most whales, but stricter regulations can apply in cases of threatened species. For example, a distance of 400 m should be kept when encountering endangered cetaceans in the St. Lawrence estuary and Saguenay river (QC) or killer whales in southern BC coastal waters. Thus, some respondents might refer to these extensions of the rules instead of the general ones.

Operators' preferred strategies to lessen underwater noise impacts on marine fauna included some speculative options (e.g., "diminish the duration of the whale-watching trips"), and some previously-implemented options - e.g., the voluntary decrease in vessel speed proposed in the framework of the Enhancing Cetacean Habitat and Observation (ECHO)

program in the Vancouver port and Cabot Strait [62, 63]. This solution was often chosen by participants working in the BC and NS areas. In these provinces, voluntary slow-down programs are implemented to protect whales from ship strikes and noise since both areas host endangered mammals (e.g., Southern Right Killer Whale, North Atlantic Right Whale). In Canada, apart from the above-mentioned underwater noise regulations enforced through the DFO, there are additional guidelines such as *Be Whale Wise* [64]. The *Be Whale Wise* goal is to educate about codes of conduct and best practices around the whale-watching experience; such best practices were mentioned on many operators' websites (with sections called "code of conduct", **Table 5**). Implementing management strategies, such as voluntary efforts to employ codes of conduct, can help promote sustainable practices. Such management strategies should be based on scientific data and evidence on operators' point of view as well as marine mammals' behavior [65–67]. Moore et al. [65] found that increased education and restrictions in management practices could help reduce disturbance to these animals, enhancing sustainable practices. For example, [68] showed that in Cape Breton Island, a tourist area in Nova Scotia, vessel impacts on long-finned pilot whales (*Globicephala melas*) foraging behaviors could be reduced by preserving foraging hotspots.

## 4.2 Web communication and outreach: The sustainable communication index as a way to enhance whale-watching tourism

Given our aim to develop a method based on data readily available to the broader public, we included in the analysis only companies with a website, while small local companies without websites were not included. Tourists' expectations of the whale-watching experience might be influenced by information displayed on websites, such as pictures of whales breaching very close to the tourists or an advertised guarantee for sightings [10, 67]. For example, tourists in Reykjavík, Iceland valued the company's "Internet presence" as one of the most important factors in their choice [69]. However, younger tourists (for instance in [69] the average age was 38 years) might be more comfortable looking for information online. Not all tourists looking for a whale-watching tour will look for options online, and small companies might advertise their activities only on the docks.

The partial awareness of the wide range of impacts of whale watching (section 3.1 and 4.1) is also reflected in the website content analysis. Web communication quality (defined in terms of completeness, correctness, and accuracy of website content) was found to be poor in Italian tourist ports, underlying similar issues [37]. By defining a Sustainable Communication Index (SCI), we propose a method to estimate the presence of specific information that makes the content complete, accurate, and correct for the whale-watching Canadian context (section 2.2). In fact, few websites mentioned whale-watching impacts (**Table 5**) and minimum distance to be kept when encountering a whale (or an explanation of its importance). A low number of mentions does not necessarily imply a non-virtuous behavior but could mean that the information is considered irrelevant or is too obvious. However, details on marine ecosystems are deemed interesting for tourists by most respondents, **Table 4B**). Analyzing operators' websites in combination with their opinions on whale watching impacts and sustainability was not possible with the data presented here, because anonymity was kept for all participants. A combined analysis would provide additional information on how they relate and support the identification of positive relations between information provided and positive behaviors. Additional research is required in order to assess the quality of tourism websites' content and define specific guidelines on the information to be included: the SCI could be generalized for other areas and/or similar tourism activities, and the website analysis could be extended from text and images to other content sources (e.g., comments, social media profiles). In the

presence of a large number of websites, it would benefit from an automated approach (e.g., using content scraping and image processing).

## 4.3 Towards a more sustainable whale watching: A continuous training framework

Whale watching can potentially be used as a tool for fostering cetacean conservation [44, 70] and advance public knowledge of cetaceans and marine conservation [46]. Our results highlighted that operators do not have a comprehensive view of the impacts their activities might impose on the marine environment, and studied websites did not display information regarding impacts on marine fauna. As part of our study, surveyed experts believe that increasing education and outreach activities directed to whale-watching operators could help improve their awareness regarding impacts and even motivate operators to reduce them [71] (section 3.3). The experts mentioned that aware whale-watching operators would be motivated to help manage tourist expectations, supporting the role of operators in enhancing tourists' knowledge of sustainable tourism and environmental impacts during trips [41]. Outreach activities should be performed with emphasis on the less perceived topics (such as biofouling and chemical pollution).

The role of education and outreach activities is widely recognized in academia, yet no previous work has pointed out the importance of outreach activities for operators. Outreach activities should be accompanied by surveys and data collection (e.g., data on the routes taken) to assess the activity's degree of sustainability. The framework of the Orams model [45], initially developed to model tourists' education, was here adjusted for whale-watching operators' education, participatory and active training. We modified the Orams model by introducing a step in the training program denominated *Impacts awareness* (blue box, **Fig 2**) as part of *The Affective Domain* and *Curiosity*. We believe that knowing impacts of tourist activities on whales might trigger positive actions towards more sustainable tourism while engaging operators and building capacity. Furthermore, the whale-watching tourist sector has the potential to initiate discussions on the role marine mammals have in terms of adaptation to climate change [72–74], and enhance involvement in conservation efforts [68, 70]. To present this view, we propose a framework that underlines the role of the scientific community before, during, and after the whale watching season (**Fig 3**). This framework supports the definition and dynamic improvement of the education and training programs envisaged by the Orams modified framework (*Design of Program* in **Fig 2**). Prior to whale-watching season, operators should be trained and provided with trusted scientific sources. During the season, the scientific community can contribute to collecting information on the operator's awareness of whale-watching impacts. At the end of the season, the feedback collected from tourists and operators should support the identification of knowledge and awareness gaps. This approach enables a continuous training loop to improve whale-watching activities for the following season.

In this study, we have identified some potential improvements and suggest linkages among the expert-operator-tourist chain that could pave the way for a long-term and synergistic improvement in whale-watching tourism. The novelty of our approach consists in targeting also whale-watching operators and not only tourists and involving scientists in several steps of the approach (e.g., *Pre-trip* and *After-trip phases*, **Fig 3**). In accordance with Tepsich et al. [42], we believe that operators should necessarily be involved, starting from a better display of information on the websites regarding the ecosystem they are going to explore during the offered tours. Our approach can contribute to developing sustainable whale-watching tourism offers. The analysis of operators' opinions and website content pointed out the aspects that need improvement (e.g., mention of whale watching impacts) and those already being mentioned (thus, in principle, followed) by most operators (e.g., minimum distance from whales). This

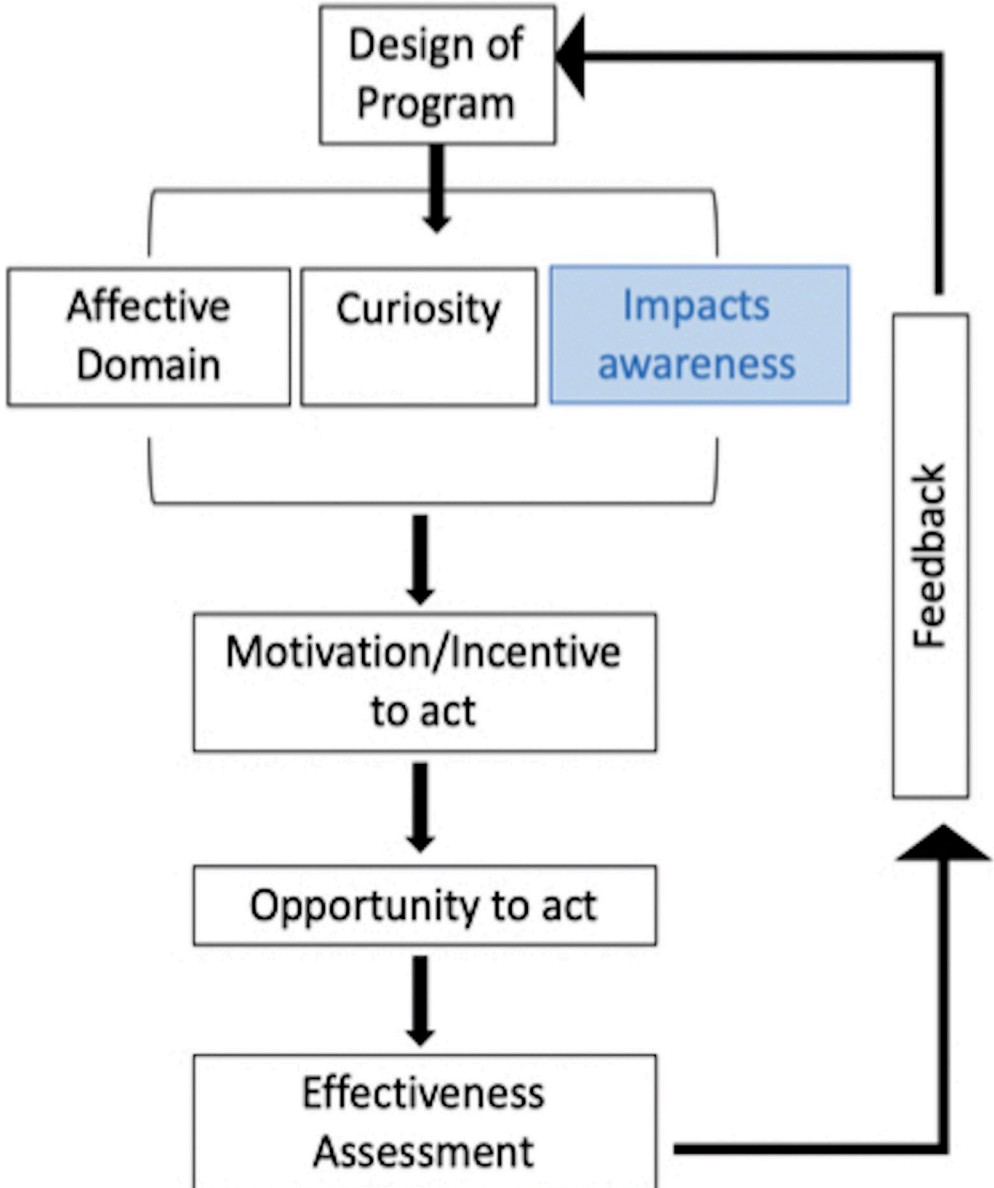

**Fig 2. Orams models [45] modified to target both tourists and whale-watching operators.** The blue box represents the proposed modification to include impact awareness as an aspect to be covered when designing education programs.

approach identifies relevant issues and aspects to be further discussed by tourist operators, policymakers, and other stakeholders and could be applied to other whale-watching locations. The involvement of operators and other stakeholders in an open, collaborative tourist experience -in order to discuss impacts and co-define solutions such as mitigation strategies- can foster pro-environmental behavior following the whale-watching activity [47]. In particular, future work should be devoted to the definition of a coherent message to the public [26] involving a multidisciplinary team of experts, scientists, and potential knowledge users such as operators and tourists (e.g., [40]). Future studies should aim at the direct involvement of

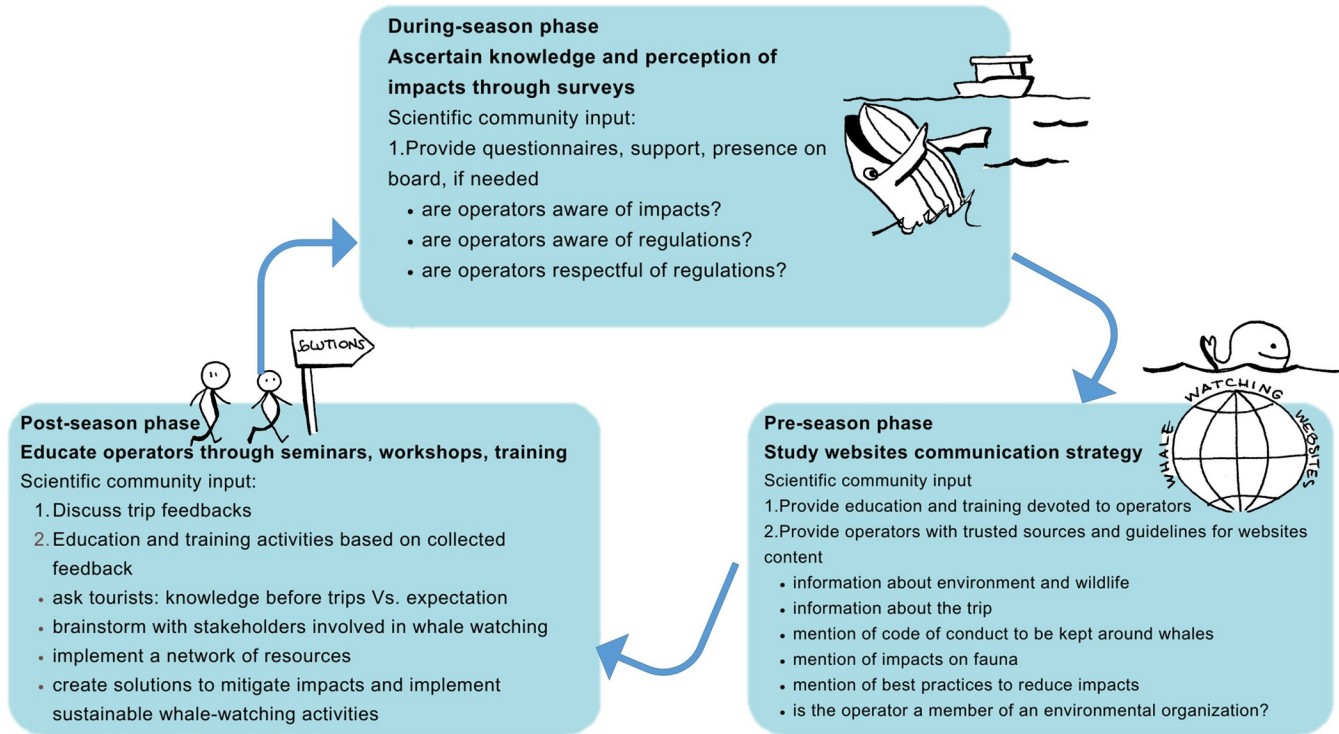

**Fig 3. Framework that underlines the contribution of the scientific community in each phase of the whale-watching activity.** Scientists can provide education and training (Pre-season) and support the collection of feedback (During-season) which contributes to dynamic improvement of the education and training programs (Post-season) in a continuous loop.

whale-watching companies and refer to their view and communication strategy, as already proposed in farming systems [75].

## 5. Conclusions

This work proposes a method to collect the perspectives of whale–watching operators regarding the possible effects of their activity on marine fauna. A questionnaire allowed to identify operators' perceptions of impacts on marine fauna and their preferred mitigation strategies. All respondents observe the current regulation, but the operators only partially perceive the wider range of impacts of the vessels on marine fauna. Few whale-watching websites mention the best practices that should be observed during whale-watching. To evaluate website content, we developed a Sustainable Communication Index (SCI) that showed that in some areas, e.g., British Columbia, websites provided more information on marine life and codes of conduct. Three whale-watching experts agreed that communication and outreach activities targeted at whale-watching operators would increase their awareness and willingness to mitigate impacts and help them manage tourists' expectations. To move in this direction, we highlight the need for continuous training and outreach programs for operators, proposing a framework for whale-watching operators´ education and training that explicitly includes the impacts of their activity on marine ecosystems. We highlight that involving whale-watching operators in communication and outreach activities would support more sustainable marine tourism.

## Supporting information

**S1 Table. Whale-watching operators' questionnaire.** Questionnaire for whale-watching operators used in this paper.
(PDF)

**S2 Table. Whale-watching experts' questionnaire.** Whale-watching experts' questionnaire. Questions between both questionnaires are linked by their identifier (ID). Questions marked with an asterisk refer to results shown to experts.
(PDF)

**S3 Table. Experts opinion on most important aspects of whale-watching trips.** Expert opinions on the most important aspects for tourists during whale-watching trips. The options selected by 2 out of 3 experts are shown in bold, while the options selected by one expert are shown in normal text.
(PDF)

**S1 Text. Whale-watching operators' questionnaire (Google form).**
(PDF)

**S2 Text. Whale-watching experts' questionnaire (Google form).**
(PDF)

**S1 Fig. Mention of impacts, best practices and distance.** Main results from the whale-watching companies' website analysis. British Columbia (BC) is shown on the left column, Nova Scotia (NS) in the middle, and the whole dataset for Canada on the right. (A) Mention of impacts on marine fauna. In the "Canada" column, results for Nova Scotia (NS), Prince Edward Island (PE), and Nunavut (NU) are not displayed. (B) Mention of best practices. In the "Canada" column, results for Prince Edward Island (PE) and Nunavut (NU) are not displayed. (C) Mention of distance kept from mammals. In the "Canada" column, results for Manitoba (MB) and Quebec (QC) are not displayed. For the whole Figure:
NL = Newfoundland and Labrador, BC = British Columbia, MB = Manitoba, NB = New Brunswick, QC = Quebec, MB = Manitoba and NS = Nova Scotia.
(PDF)

## Acknowledgments

Elena Zwirner (Université Clermont Auvergne) is gratefully acknowledged for her advice on questionnaire guidelines. We also thank whale-watching tour operators who completed the questionnaire. We greatly appreciate Eric Hoyt (Research Fellow, Whale and Dolphin Conservation), Ted Cheeseman (Expedition leader, Cheesemans' Ecology Safaris; Co-Founder & Director, Happywhale), and Heidi Pearson (Associate Professor of Marine Biology, University of Alaska Southeast) for dedicating their time in participating in the experts' questionnaire. We also thank Jay Frentress for his constructive review and language editing which improved the quality of the manuscript.

## Author Contributions

**Conceptualization:** Alice Affatati, Chiara Scaini, Anna Scaini.

**Data curation:** Alice Affatati.

**Formal analysis:** Alice Affatati.

**Funding acquisition:** Anna Scaini.

**Investigation:** Alice Affatati.

**Methodology:** Alice Affatati, Chiara Scaini, Anna Scaini.

**Validation:** Chiara Scaini.

**Visualization:** Alice Affatati.

**Writing – original draft:** Alice Affatati.

**Writing – review & editing:** Alice Affatati, Chiara Scaini, Anna Scaini.

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
