## [Decision Letter · Decision Letter 0]

13 Mar 2023

PONE-D-22-32212Enhancing sustainable whale-watching tourism: opinions and potential role of operatorsPLOS ONE

Dear Dr. Scaini,

Thank you for submitting your manuscript to PLOS ONE. After careful consideration, we feel that it has merit but does not fully meet PLOS ONE’s publication criteria as it currently stands. Therefore, we invite you to submit a revised version of the manuscript that addresses the points raised during the review process.

I agree with the points raised by the two reviewers. I would encourage the authors to ensure that they address the points made by the reviewers. In particular, I would expect the authors to recognise and acknowledge the significant limitations of this research, particularly in terms of your data collection and analysis.Both reviewers have included references for a significant amount of relevant literature that should be carefully considered and addressed in your work. This will likely improve setting the context in which your work sits, as well as your discussion. (As both reviewers note, your context and motivation is rather weak).I would recommend that you take the time to address the coherence of the manuscript once all of the other amendments are complete. I would recommend the use of Joseph Williams' "Style: Lessons in Clarity and Grace" to assist in improving the coherence.

We look forward to receiving your revised manuscript.

Kind regards,

Peter Edwards

Academic Editor

PLOS ONE

Journal Requirements:

2. You indicated that ethical approval was not necessary for your study. We understand that the framework for ethical oversight requirements for studies of this type may differ depending on the setting and we would appreciate some further clarification regarding your research. Could you please provide further details on why your study is exempt from the need for approval and confirmation from your institutional review board or research ethics committee (e.g., in the form of a letter or email correspondence) that ethics review was not necessary for this study? Please include a copy of the correspondence as an ""Other"" file.

4. Please include a complete copy of PLOS’ questionnaire on inclusivity in global research in your revised manuscript. Our policy for research in this area aims to improve transparency in the reporting of research performed outside of researchers’ own country or community. The policy applies to researchers who have travelled to a different country to conduct research, research with Indigenous populations or their lands, and research on cultural artefacts. The questionnaire can also be requested at the journal’s discretion for any other submissions, even if these conditions are not met.  Please find more information on the policy and a link to download a blank copy of the questionnaire here: https://journals.plos.org/plosone/s/best-practices-in-research-reporting. Please upload a completed version of your questionnaire as Supporting Information when you resubmit your manuscript.

AA was funded by OGS Project BlueSkills: Blue Jobs and Responsible Growth in the Mediterranean, labeled by the UfM and funded by the Italian Ministry of University and Research (MUR) 

AS was funded by the Department of Physical Geography, Stockholm University. Stockholm University supports open access publishing by covering article-processing charges.

Funding for this study has been provided by OGS Project BlueSkills: Blue Jobs and Responsible Growth in the Mediterranean, labeled by the UfM and funded by the Italian Ministry of University and Research (MUR) and by the Department of Physical Geography, Stockholm University. 

However, funding information should not appear in the Acknowledgments section or other areas of your manuscript. We will only publish funding information present in the Funding Statement section of the online submission form. 

AA was funded by OGS Project BlueSkills: Blue Jobs and Responsible Growth in the Mediterranean, labeled by the UfM and funded by the Italian Ministry of University and Research (MUR) 

7. Your ethics statement should only appear in the Methods section of your manuscript. If your ethics statement is written in any section besides the Methods, please move it to the Methods section and delete it from any other section. Please ensure that your ethics statement is included in your manuscript, as the ethics statement entered into the online submission form will not be published alongside your manuscript. 

8. We note that Figure 1 in your submission contain map images which may be copyrighted. All PLOS content is published under the Creative Commons Attribution License (CC BY 4.0), which means that the manuscript, images, and Supporting Information files will be freely available online, and any third party is permitted to access, download, copy, distribute, and use these materials in any way, even commercially, with proper attribution. For these reasons, we cannot publish previously copyrighted maps or satellite images created using proprietary data, such as Google software (Google Maps, Street View, and Earth). For more information, see our copyright guidelines: http://journals.plos.org/plosone/s/licenses-and-copyright.

Reviewers' comments:

Reviewer's Responses to Questions

**Comments to the Author**

1. Is the manuscript technically sound, and do the data support the conclusions?

Reviewer #1: Yes

Reviewer #2: Partly

2. Has the statistical analysis been performed appropriately and rigorously? 

Reviewer #1: N/A

Reviewer #2: Yes

3. Have the authors made all data underlying the findings in their manuscript fully available?

Reviewer #1: Yes

Reviewer #2: Yes

4. Is the manuscript presented in an intelligible fashion and written in standard English?

Reviewer #1: Yes

Reviewer #2: Yes

5. Review Comments to the Author

Reviewer #1: General comments:

The manuscript at hand provides an interesting perspective on sustainability issues of whale-watching tourism by focusing on the tour operators and their websites. The research is mostly based on two online surveys with mostly closed questions. I personally think that qualitative, semi-structured interviews would have been much more appropriate compared to a questionnaire with closed questions sent to only three respondents in the case of the whale watching experts, but also in case of the operators. Such personal interviews would provide a much higher level of insights and reflection. Being aware that the authors cannot change their research design ex-post I suggest that the authors at least recognize these aspects as limitations of their study (which are completely missing at the moment).

Would it be possible to link the tour operators’ answers to the website analysis? For instance, those who indicate higher importance of sustainability might also foster these issues more on their websites? (cross-tabulation)

In the introduction: Please stress the innovative character of this research more strongly and please highlight the research gap more clearly.

Detailed comments:

Abstract

Page 1, Line 35: “to develop a framework on operators' perception” => of operators' perception

1. Introduction

P2, Line 47: “crucial for human health” => not only!

P2, L48: “Wildlife tourism is a specialized tourism activity” => Marine wildlife… tourism form

P2, L51: “Hoyt [11] defined whale watching” => Please add page number for quotations, thanks

P2, L54: “been considered ecotourism because” => as an ecotourism activity

P2, L56: “as analyzed by O’Connor” => as analyzed by O’Connor et al.

P2, L57: “excursions might affect” => might negatively affect?

P2, L61: “whale-watching is harmful to whales” => potentially harmful? or could be harmful

P3, L66-68: Is this vague definition necessary here?

P3, L78: “Online websites are pivotal” => Wording. Are there also offline websites?

P3, L79: “becoming increasingly relevant” => relevant for what?

P4, L93: “most studies have been conducted” => studies about what?

P4, L97: “no studies have been developed specifically for whale-watching operators“ => Really? I cannot believe that, honestly. However, there are studies intensively analyzing WW operators perspectives, for instance Mayer et al., 2018, https://doi.org/10.1016/j.ocecoaman.2018.04.016

P4, L102-103: Use these information as reason for choosing Canada as study area? In L98 the readers might wonder why exactly you choose Canada.

P4, L105: “found that in the Salish Sea whales,” => Unclear, please explain better, thanks.

P4, L108: “with a range of qualitative methods” => Is a questionnaire with closed questions a qualitative method?

P4, L113: “touristic activity“ => tourism, see also L117, L153, L393

2. Materials and Methods

L146: Unnecessary in my opinion

Table 1, ID 1: How did you come up with these numbers here?

ID 2: Why not an increasing scale?

ID3-ID5: Multiple answers possible? Please always indicate the relevant n later.

L154: “Based on their work” => Whose work?

L156: “the following information was ranked” => assessed: Yes/No is no ranking in my opinion

L163: “Does the company respect the required distance“ => seemingly? or according to their website?

L163f.: “(note that it should be at least 100 m)?“ => According to whom? Reference?

L170f.: “The normalized SCI of the set of considered sites was computed by dividing the number of total websites searched by the SCI“ => Sorry, I don't understand this analysis step

L175:” with Dr. Enric Hoyt” => Typo. Eric.

L191: “The complete list of questions and options is shown in Table 2.“ => Could be in the appendix/online supplement; not really necessary here

3. Results

L200: “The number of responders” => respondents

L207f.: the overall sample size is not very large, but you reduce it further with this selection. Why should the provincial origin of the respondents influence the answers?

L217f.: social desirability?

L220-225: Such results are better presented in a table. The current version is a bit tiring to read.

L223: “in BC stated that their vessel stays 100 m far” => This response does not fit to the question "should be away"...

L230: “we can see that perception” => the perception

Table 3: What is the exact difference between the last two lines? Please explain, thanks.

4. Discussion

L319: Please provide more context: What kind of study is this? A whale watcher survey?

L346: claimed to be followed! Do you expect operators not abiding to the rules to indicate this in a questionnaire?

L367: “Their findings showed that increased education” => of whom? the operators?

L370: “long‐ finned pilot whales” => long-finned

L424: “website's analysis“ => website analysis

L432f.: plural form: views and strategies

L446: “the whale-watching tourist” => tourism

Figure 2: Add n please

Figure 3: Add BC here and NS below please

Figure 4: Readbility... Again: add BC and NS

Figure 5: Please add the n here

Figure 8: Layout not very compelling => empty spaces missing

"-)" does not look good

References:

Some critical basic literature about the contested sustainable nature of WW is missing:

Higham, J., Bejder, L., Allen, S. J., Corkeron, P. J., & Lusseau, D. (2016). Managing whale-watching as a non-lethal consumptive activity. Journal of Sustainable Tourism, 24(1), 73–90. https://doi.org/10.1080/09669582.2015.1062020

Heenehan, H., Basurtoa, X., Bejder, L., Tyne, J., Higham, J.E.S., Johnston, D.W., 2015. Using Ostrom's common-pool resource theory to build toward an integrated ecosystem-based sustainable cetacean tourism system in Hawaii. J. Sustain Tour. 23 (4), 536–556. https://doi.org/10.1080/09669582.2014.986490.

Mayer et al. 2008 included WW tour operators in their qualitative survey of WW stakeholders in Baja California, Mexico

Mayer, M., Brenner, L., Schauss, B., Stadler, C., Arnegger, J., & Job, H. (2018). The nexus between governance and the economic impact of whale-watching. The case of the coastal lagoons in the El Vizcaíno Biosphere Reserve, Baja California, Mexico. Ocean and Coastal Management, 162, 46–59. https://doi.org/10.1016/j.ocecoaman.2018.04.016

In this study, you could get additional evidence for the importance of websites for choosing WW tour operators (see Table 2):

Lissner, I. & Mayer, M. (2020) Tourists’ willingness to pay for Blue Flag's new ecolabel for sustainable boating: the case of whale-watching in Iceland, Scandinavian Journal of Hospitality and Tourism, 20:4, 352-375, DOI: 10.1080/15022250.2020.1779806

[2] Higgins-Desbiolles F. The “war over tourism”: challenges to sustainable tourism in the tourism academy after COVID-19. Journal of Sustainable Tourism, 2000; 29(4), 551-569.

https://doi.org/10.1080/09669582.2020.1803334 => Surely not published in 2000 but rather in 2020?

[4] Rutty M, Gössling S, Scott D, Hall CM. The global effects and impacts of tourism: an overview. The Routledge handbook of tourism and sustainability, 2005; 54-82. =>Editors? Publisher?

[11] Hoyt E. Whale watching 2001: Worldwide tourism numbers, expenditures, and expanding socioeconomic benefits. 2001 => publisher?

[27] Parsons ECM. The negative impacts of whale-watching. Journal of Marine Biology, 2012. https://doi.org/10.1155/2012/807294 => Article number? Or Volume/Issue/Pages?

[45] Moscardo G, Saltzer R. Understanding tourism wildlife interactions. Sustainable Tourism Cooperative Research, 2005; 36. => A journal or an edited volume?

Reviewer #2: This manuscript aims to understand operators’ perceptions about some mitigation actions to reduce the impacts of the activity on cetaceans, also contrasting the evidence with three experts in the field. This is particularly important for the whale-watching tourism sector since, as the authors underline, there is still scant research concerning operators’ opinions.

Overall, I would like to invite the authors to rethink the manuscript and be more ambitious, i.e., I encourage them to reflect on where to put more focus and emphasis. It might be a reasonable contribution to the field. However, I'll suggest some improvements and ask the authors for a major paper revision.

Title

In my opinion, the title is pretty general, although the work has the potential to be summarised in a more attractive and engaged title. I invite the author to consider re-write it.

Abstract

After considering the suggestions, I also invite the authors to re-do it.

Introduction

As you know, the introduction is one of the most important sections of a manuscript, motivating your research. That is why I encourage the authors to revise the introduction and provide some more order and conciseness (see, e.g., the third paragraph).

The authors should be consistent with the terminology they use around tourism. In the abstract, you talk about “green tourism”. At the same time, you define whale-watching as wildlife-based tourism considered a form of ecotourism since it should adopt the sustainable use of cetaceans (a definition which is definitely right). On the other hand, in the fourth paragraph, you jump into coastal and marine tourism. Please, make it clearer.

Still about the fourth paragraph (lines 75-88). You cite Olszewski-Strzyzowski [31] to talk about some existing good practices for promoting sustainability. However, I think there are ample examples in the whale-watching literature focusing on this issue that you are not citing. Please, reconsider doing a more specific literature review in this regard (overall, along with all the manuscript).

Likewise, you begin to talk about online websites straightforwardly. My recommendation is to introduce this point by talking about the importance of how communication impacts consumer expectations and projections about the activity. Please, see the following articles, especially the first one, which is in line with one of your research aims.

- Judge, C., Penry, G. S., Brown, M., & Witteveen, M. (2020). Clear waters: assessing regulation transparency of website advertising in South Africa’s boat-based whale-watching industry.

- Bertella, G. (2019). Close encounters with wild cetaceans: Good practices and online discussions of critical episodes.

- Finkler, W., & Higham, J. E. (2020). Stakeholder perspectives on sustainable whale watching: A science communication approach.

With regard to the fifth paragraph, please see the following articles, focusing on operators:

- Curtin, S. (2010). Managing the wildlife tourism experience: The importance of tour leaders.

o Despite it is quite old, it should be recommendable to read it, also suitable for increasing the background about operators’ role.

- Hoarau, H., & Kline, C. (2014). Science and industry: Sharing knowledge for innovation.

- Hoarau-Heemstra, H., & Eide, D. (2019). Values and concern: Drivers of innovation in experience-based tourism.

In addition, what do you refer with “categories of whale-watching”? I think you are talking about the attributes or aspects that define the experience and/or complement it. In that sense, I invite you to read the following publications about IPA (similar to Tepsich et al.), which would probably help you provide further information and some examples in this regard.

- Bentz, J., Lopes, F., Calado, H., & Dearden, P. (2016). Enhancing satisfaction and sustainable management: Whale watching in the Azores.

- Suárez-Rojas, C., Hernández, M. G., & León, C. J. (2023). Segmented importance-performance analysis in whale-watching: Reconciling ocean coastal tourism with whale preservation.

I also suggest you summarise the next paragraph (lines 98-107) and complement, in that case, the methodology section. Finally, concerning the last three research questions/ aspects, I invite you to emphasise the last one (see my comments in the discussion)

Methodology

Section 2.1. Please, I would like you to provide some additional information about the operators. Did you approach them only once? Or did you send them different rounds of emails to remember/ encourage completing the survey? On the other hand, did you pre-test the questionnaire or organise a focus group to validate it? And the last question, is your final sample representative of the target population (Canadian whale-watching operators)?

Section 2.2. Put more emphasis on the Sustainable Communication Index. It is interesting!

At some point, in the introduction or the methodology, I’m missing some specific paragraph or section explaining the impacts you include in the questionnaire (short literature review). Suppose you compare maritime transport vessels and whale-watching ones. Do you think the issues you are citing are representative impacts or problems to be prioritised in the context of whale-watching (e.g., the introduction of invasive species)? In this regard, the way you formulated the question (ID 3), in my opinion, could be being misunderstanding. I mean, if you are referring to ships overall, these are really significant impacts; however, are with respect to whale-watching vessels too? Following the example of invasive species, I do not consider whale-watching ships critically contributing to this issue (they do not navigate such long distances and transit through different oceans). That is why I encourage you to explain somewhere the impacts you are studying.

Results

Section 3.1. Following my aforementioned comments, why do you think “all of them” was the correct option to be selected? (lines 226-227).

Section 3.2. Could be any possibility to relate the operator’s opinion with the information they provide on their websites? That is, could you confirm if there is any positive relation between those websites that mention aspects about the distance and other sustainable questions with those operators who are more concerned about the whale-watching impacts or the strategies to face them? This would be interesting to know.

Discussion

Section 4.1. Lines 324-332. I encourage you to look for some papers in whale-watching literature that have segmented whale-watching consumers, identifying different profiles, from those who effectively expect close encounters to those who are more environmentally friendly.

Line 337-338. From operators’ responses, are you sure you could affirm that they are overall low-aware? As you later underline from experts’ opinions, it could probably be in relation to (the lack of) information. Please, also consider my previous comments about the impacts.

Lines 349-351. It would be interesting if you were able to provide an in-depth argument and relate these regulations with the specific areas of the responding operators (and even operators’ answers).

Section 4.2. 2nd paragraph (lines 380 – 394). Please, check this paragraph. I consider some information quite repetitive.

On the other hand, please revise lines 395-405. It is quite confusing the way you are presenting the discussion. Some questions look as if they belonged to the previous section.

Lines 409-413. Have you found any evidence in this or another context that relates the information provided on the websites to consumer satisfaction? Are both forms (website and onboard) of providing information about whale biology, regulations, etc., a precondition for a good whale-watching experience, or complementary? It would be interesting if you could discuss something in this regard.

Section 4.3. To what method are you referring in the title? My opinion in this section is that you are providing an interesting contribution towards further directions in whale-watching (Fig 7). However, although it has potential, it is not clearly presented, and therefore, its originality is not easily identified. My suggestion is to refocus the paper around this method, being the empirical results evidence that motivates it.

Conclusions.

You do not cite here the contribution of section 4.3, and you could reconsider that. Overall, it would be best if you were more specific about your (theoretical and empirical) contributions and future directions for the sector.

Finally, what are the limitations of this study?

All the best with your revisions!

6. PLOS authors have the option to publish the peer review history of their article (what does this mean?). If published, this will include your full peer review and any attached files.

Reviewer #1: No

Reviewer #2: No

---

## [Author Response · Author response to Decision Letter 0]

30 Jun 2023

PONE-D-22-32212

The role of operators in sustainable whale-watching tourism: proposing a continuous training framework 

Response to reviewers

The referees’ comments are shown in black, plain text, with our responses embedded in blue and italics (in the attached word file).

Reviewer #1: 

General comments:

The manuscript at hand provides an interesting perspective on sustainability issues of whale-watching tourism by focusing on the tour operators and their websites. The research is mostly based on two online surveys with mostly closed questions. I personally think that qualitative, semi-structured interviews would have been much more appropriate compared to a questionnaire with closed questions sent to only three respondents in the case of the whale watching experts, but also in case of the operators. Such personal interviews would provide a much higher level of insights and reflection. Being aware that the authors cannot change their research design ex-post I suggest that the authors at least recognize these aspects as limitations of their study (which are completely missing at the moment).

We thank the reviewer for their insights. The research was set up during the COVID-19 pandemic lockdown. Therefore, online questionnaires were the only possible option. Moreover, in order to perform in-person interviews, we would have needed funding to travel for this project. 

We have added some of the limitations related to the choice of performing questionnaires in the discussion to reflect on the methodology and on how it could be best to perform the study using interviews whenever possible. Some advantages, such as accessibility, time and cost efficiency, flexibility, interactivity without interviewer bias, unrestricted geographic coverage are highlighted in the paper Chang, T. Z. D., & Vowles, N. 2013. Strategies for improving data reliability for online surveys: A case study. International Journal of Electronic Commerce Studies, 4(1), 121-130.).

In the methodology, we have included a sentence to highlight the semi-structured interviews as a potentially more suited methodology in the case of the three experts (lines 191-193):

“More personal, semi-structured interviews with the three experts could provide a higher level of insights and reflection, which was only partly achieved using free-text options”.

We are aware of the high potential of interviews with respect to online questionnaires and we hope to carry out them during future work.

Would it be possible to link the tour operators’ answers to the website analysis? For instance, those who indicate higher importance of sustainability might also foster these issues more on their websites? (cross-tabulation)

We initially discussed this possibility within the working group, but because of the anonymity of the questionnaire responses, there is no way to connect the two types of information. We hope to analyze the relation between operators’ point of view and website communication strategies in the future.

In the introduction: Please stress the innovative character of this research more strongly and please highlight the research gap more clearly.

We have taken into account this point and present a revised introduction highlighting the research gap and the novelty of the study. As a result, the introduction has been largely rewritten. The introduction is now shorter, and each paragraph details a challenging aspect of whale-watching tourism. The last paragraph was also improved by adding more clearly the novelty of the approach and focusing on the continuous training, e.g., (lines 70-81):

“Here, we address the point of view of whale-watching tourist companies regarding interactions between vessels and marine fauna and possible solutions to mitigate impacts. This study investigates and evaluates whale-watching operators´ perception of the impacts on marine fauna related to their activities, and proposes a framework for continuous training devoted to whale-watching operators and involving scientific knowledge. A novel compound approach consisting of a combination of questionnaires and data obtained from website analysis is used to tackle the following aspects:

1. what whale-watching operators know about the acoustic impact of their activity on marine mammals, 

2. how whale-watching operators communicate through their websites regarding tourism activity and its impacts on the marine ecosystem,

3. strategies to enhance the sustainability of whale-watching tourism with selected experts in the field”.

Detailed comments:

Abstract 

Page 1, Line 35: “to develop a framework on operators' perception” => of operators' perception

The sentence was modified as suggested.

1. Introduction 

P2, Line 47: “crucial for human health” => not only!

We added “marine fauna” to broaden the meaning of the sentence, as pointed out by the reviewer.

P2, L48: “Wildlife tourism is a specialized tourism activity” => Marine wildlife… tourism form

The sentence has been erased for clarity, and to avoid using too many terms related to sustainable tourism 

P2, L51: “Hoyt [11] defined whale watching” => Please add page number for quotations, thanks

Page number was added as suggested.

P2, L54: “been considered ecotourism because” => as an ecotourism activity

“As an ecotourism activity” was added.

P2, L56: “as analyzed by O’Connor” => as analyzed by O’Connor et al.

O’Connor et al. was substituted by the reference number

P2, L57: “excursions might affect” => might negatively affect?

We have changed the sentence to “potentially affect” 

P2, L61: “whale-watching is harmful to whales” => potentially harmful? or could be harmful

We have modified the sentence with “could be harmful”

P3, L66-68: Is this vague definition necessary here?

The sentence was erased to follow the suggestion

P3, L78: “Online websites are pivotal” => Wording. Are there also offline websites?

“Online” was erased since it was not necessary, as suggested

P3, L79: “becoming increasingly relevant” => relevant for what?

This sentence was rewritten as “According to [30], websites are pivotal for promoting tourism nature-related activities, as also shown for European countries by [34]. “

P4, L93: “most studies have been conducted” => studies about what?

“Whale watching” was added to explain the type of studies

P4, L97: “no studies have been developed specifically for whale-watching operators“ => Really? I cannot believe that, honestly. However, there are studies intensively analyzing WW operators perspectives, for instance Mayer et al., 2018, https://doi.org/10.1016/j.ocecoaman.2018.04.016

P4, L102-103: Use these information as reason for choosing Canada as study area? In L98 the readers might wonder why exactly you choose Canada.

The sentence was made clearer. It is now included as (lines 82-85):

“The analysis focuses on Canada at a national scale and selected provinces characterized by different degrees of tourism development and the presence of different marine species, including endangered ones. Since the 1990s, the presence of 30 whale species has led to the growth of the whale-watching industry in Canada”.

P4, L105: “found that in the Salish Sea whales,” => Unclear, please explain better, thanks.

Following this comment and to enhance clarity, this part was deleted and the sentence rephrased to keep the reference (Frayne et al., 2020) (lines 85-87).

“However, whale populations are depleted due to anthropogenic stressors, e.g., pollution [49]; therefore, the Canadian government has strengthened its regulations to protect cetaceans and increase tour operators’ awareness towards potential impacts [50]”. 

P4, L108: “with a range of qualitative methods” => Is a questionnaire with closed questions a qualitative method?

This comment raised an important point. “Qualitative” was deleted from the sentence to avoid confusion.

P4, L113: “touristic activity“ => tourism, see also L117, L153, L393

“Touristic activity” was replaced by “tourism”. All the suggestions have been taken into consideration in the revised version of the manuscript.

2. Materials and Methods 

L146: Unnecessary in my opinion

We added questions identifiers (IDs) in order to keep a tidier manuscript. Since we are referring to many questions (e.g., whale-watching operators' questionnaire, experts’ questionnaire), we believe that adding IDs will make it easier for the readers to keep track of the discussed topics.

Table 1, ID 1: How did you come up with these numbers here?

We selected 100 and 200 m because Canada's Marine Mammal Regulations state “Keeping a minimum of 100 meters away from most whales, dolphins, and porpoises, and keeping a minimum of 200 meters away if they are in resting position or with their calf.” We chose to use only multiple questions and not give the participants the possibility to write their own preference, so we picked other reasonable numbers to evaluate tour operators' perception. We selected values in the upper and lower bound of the prescribed distance (displaying them in the question in an increasing order) in order to check if the operators knew the limitations, and to partially prevent biased responses.

ID 2: Why not an increasing scale?

We did not choose an increasing scale to try to avoid bias and discourage respondents towards selecting a random number. 

ID3-ID5: Multiple answers possible? Please always indicate the relevant n later.

The relevant n was added in Table 4 (which replaced Figures 2,3,4) as well as in Figure S3 and Table S4.

L154: “Based on their work” => Whose work?

The reference to “[35] Benevolo C, Spinelli R. The quality of web communication by Italian tourist ports. Tourism: An International Interdisciplinary Journal, 2018; 66(1), 52-62” was missing. The reference was added to the revised version of the manuscript.

L156: “the following information was ranked” => assessed: Yes/No is no ranking in my opinion

“Ranked” was modified to “Assessed”

L163: “Does the company respect the required distance“ => seemingly? or according to their website?

To address this comment, we changed the sentence to “Does the company declare”. 

L163f.: “(note that it should be at least 100 m)?“ => According to whom? Reference?

The reference was added as suggested.

L170f.: “The normalized SCI of the set of considered sites was computed by dividing the number of total websites searched by the SCI“ => Sorry, I don't understand this analysis step

We have improved the sentence as follows:

“The normalized SCI was computed by dividing the SCI by the number of websites inspected. The operation was done for each area (column 2 to 6) and for all websites (last column). This allows comparing the scores obtained for the different areas considered”.

We also explain why it was defined, e.g., in order for results in different areas to be comparable despite the different number of websites considered. The caption of Table 4 is now modified to: “Table 4: Factors considered for assessing SCI and rating obtained in each province (columns 2 to 6) and for all provinces (column 7). The last two rows provide absolute and normalized SCI values (obtained by dividing the absolute SCI by the number of websites inspected, provided in row 8).”

L175:” with Dr. Enric Hoyt” => Typo. Eric.

Thanks for pointing this out; the typo was fixed.

L191: “The complete list of questions and options is shown in Table 2.“ => Could be in the appendix/online supplement; not really necessary here. 

We moved the complete questionnaire to the supporting information.

3. Results 

L200: “The number of responders” => respondents

“Responders” was changed to “respondents”.

L207f.: the overall sample size is not very large, but you reduce it further with this selection. Why should the provincial origin of the respondents influence the answers?

We slightly modified the sentence so that the main reason for our choice (the larger number of respondents) is clear (lines 228-230):

“We analyze results specifically for BC and NS because they had the highest response rates in the questionnaire - 12 for BC and 5 NS, respectively (The number of respondents in the remaining provinces was 9)”. 

In the discussion, we describe the context of the two considered areas and how they might explain our findings (e.g. the communication campaigns carried out in some areas might justify higher attention to some topics), lines 349-351:

“Another factor that might influence perception is that Canada's public outreach activities are mostly devoted to acoustic pollution, with less attention to other impacts [60] “

L217f.: social desirability? 

To account for the issue of social desirability, we have added some information to explain how we tried to reduce bias. Section 2.1, lines 127-130:

“In designing the questionnaire, we have addressed social desirability bias, e.g., the tendency of respondents to bias their choices to comply with social norms. For instance, in questions ID1 and ID2 we added options with values similar, but not equal, to the correct ones [55]. “

Also, keeping anonymity should help avoiding social desirability bias. This information was added in section 2.1 (“In designing the questionnaire, we have addressed social desirability bias, e.g., the tendency of respondents to bias their choices to comply with social norms [...] The consent form is also designed to highlight subject anonymity, which is another common measure to reduce biased responses.”) and section 4.1.

L220-225: Such results are better presented in a table. The current version is a bit tiring to read.

A table with these results was added (Table 3) and the text was modified to read (lines 234-236):

“In general, we found good knowledge and observance of regulations, with most operators observing the minimum distance of 100 m. All participants operating in BC and NS answered that their vessel keeps a distance of around 100 m, or more, from the whales during tours (Table 3).”

“Table 3: Detailed results to Questions ID 1 and 2 for BC and NS participants

ID Question BC [participants; n=12] NS [participants; n=5]

ID 1 “How far does your vessel go from the whale during the tours?” • “More than 200 m”:6

• “More than 100 m”:5 

• “Around 100 m”: 1 • “More than 100 m”: 1

• “Around 100 m”: 4 

ID 2 “How far away should the vessel be from the cetacean at least?” • at least 100 m: 11

• at least 200 m: 1 • “100 m”: 5

 L223: “in BC stated that their vessel stays 100 m far” => This response does not fit to the question "should be away"...

The paragraph (lines 220-225 in the former document) was erased and the results were summarized in a table, as suggested (Table 3, see above).

L230: “we can see that perception” => the perception

The article “the” was added.

Table 3: What is the exact difference between the last two lines? Please explain, thanks.

We have improved the instances where we introduce the normalized SCI, which is reported in the last line of Table 3 (Table 4 in the new version of the manuscript). To clarify the difference between SCI and normalized SCI, we improved the definition of normalized SCI as follows (lines 184-187):

“The normalized SCI was computed by dividing the SCI by the number of websites inspected. The operation was done for each area (column 2 to 6) and for all websites (last column). This allows comparing the scores obtained for the different areas considered”.

We also explain why it was defined, e.g., in order for results in different areas to be comparable despite the different number of websites considered.

The caption of Table 4 is now modified to: “Table 4: Factors considered for assessing SCI and rating obtained in each province (columns 2 to 6) and for all provinces (column 7). The last two rows provide absolute and normalized SCI values (obtained by dividing the absolute SCI by the number of websites inspected, provided in row 8)”.

4. Discussion

L319: Please provide more context: What kind of study is this? A whale watcher survey?

A brief description has been added to Section 4, lines 302-314:

“Our work combined sources of information through perception data from questionnaires to whale-watching operators, website content and experts’ opinions. The responses collected from whale-watching operators are not statistically representative of the entire population of whale-watching operators in Canada, preventing us from conducting a statistical analysis and more comprehensive scrutiny of the results, but nonetheless provide useful insights to understand their point of view (section 4.1). The whale-watching companies´ communication strategy is then analyzed by introducing a Sustainable Communication Index (section 4.2). The questionnaire can be adapted with minimum changes to different settings and case studies, creating a pliant method that, together with the websites’ analysis, can complement the existing works on tourists’ perspectives of whale-watching activities. Further work including interviews and direct interaction with whale-watching operators is pivotal to improve operators' perceptions of overall impacts on marine mammals and move towards a continuous training framework (presented in section 4.3) supporting the development of a more sustainable tourism activity”.

L346: claimed to be followed! Do you expect operators not abiding to the rules to indicate this in a questionnaire?

Following the reviewer´s question, we have modified the sentence to make it clear that we refer to the mentions of minimum distance in websites, making reference to Table 5 where the results of the websites analysis are shown. We also postulate that those who mention the limitations are more prone to observe them (lines 353-361):

“More than half of the analyzed Canadian whale-watching websites mention existing rules regarding keeping a minimum or suggested distance from the cetaceans, which suggests that they are prone to observe them (Table 5). BC operators claim to be more conservative when approaching animals at a distance greater than 100 m compared to NS operators. The Fisheries and Oceans Canada (DFO) regulations are different in case endangered species are present in the area [61]. Vessels should keep a minimum of 100 m from most whales, but stricter regulations can apply in cases of threatened species. For example, a distance of 400 m should be kept when encountering endangered cetaceans in the St. Lawrence estuary and Saguenay river or killer whales in southern BC coastal waters. Thus, some respondents might refer to these extensions of the rules instead of the general ones”.

To further address this topic, we included a sentence in the discussion to underline the limitations associated with social desirability bias in surveys such as the one carried out in our study (lines 317-320):

“Online questionnaires were chosen to study the perception of operators because of time and cost efficiency, flexibility, interactivity without interviewer bias and geographic coverage. In addition to the low number of responses, some limitations exist regarding respondent truthfulness [57]. Moreover, results might be subjected to a degree of bias, in particular due to social desirability [55]”.

L367: “Their findings showed that increased education” => of whom? the operators?

The authors’ findings. A reference “[68] McComb‐Turbitt S, Costa J, Whitehead H, Auger‐Méthé M. Small‐scale spatial distributions of long‐finned pilot whales change over time, but foraging hot spots are consistent: Significance for marine wildlife tourism management. Marine Mammal Science, 2021; 37(4), 1196-1211. https://doi.org/10.1111/mms.12821” was added to clarify this (lines 376-379):

L370: “long‐ finned pilot whales” => long-finned

The typo was fixed. 

L424: “website's analysis“ => website analysis

We have corrected the sentence as suggested.

L432f.: plural form: views and strategies

We have corrected the sentence as suggested.

L446: “the whale-watching tourist” => tourism

We have corrected the sentence as suggested.

Figure 2: Add n please

Figure 3: Add BC here and NS below please

Figure 4: Readbility... Again: add BC and NS

Figure 5: Please add the n here

Figure 8: Layout not very compelling => empty spaces missing

"-)" does not look good

The figures were modified following the above suggestions, adding n, BC and NS, and improving editing and layout. To improve readability, Figures 2, 3 and 4 were merged and a table with a clearer outline. 

Given the substantial changes to the focus of the manuscript and the readability issues with our previous version of the figures, we have decided to show the results of the questionnaire in the form of tables. We moved some of the information relative to the questionnaire results to the appendix. In particular:

- previous versions of figures 2-3-4 have been summarized in a table (Table 4), 

- figure 5 has been moved to an appendix as supporting/additional results, 

- previous version of figure 6 has been summarized in a table and moved to an appendix as supporting/additional results, showing in bold the options that two out of 3 experts have selected, and in normal text the other option selected by one of the three experts.

References:

Some critical basic literature about the contested sustainable nature of WW is missing:

Higham, J., Bejder, L., Allen, S. J., Corkeron, P. J., & Lusseau, D. (2016). Managing whale-watching as a non-lethal consumptive activity. Journal of Sustainable Tourism, 24(1), 73–90. https://doi.org/10.1080/09669582.2015.1062020

Heenehan, H., Basurtoa, X., Bejder, L., Tyne, J., Higham, J.E.S., Johnston, D.W., 2015. Using Ostrom's common-pool resource theory to build toward an integrated ecosystem-based sustainable cetacean tourism system in Hawaii. J. Sustain Tour. 23 (4), 536–556. https://doi.org/10.1080/09669582.2014.986490.

Mayer et al. 2008 included WW tour operators in their qualitative survey of WW stakeholders in Baja California, Mexico

Mayer, M., Brenner, L., Schauss, B., Stadler, C., Arnegger, J., & Job, H. (2018). The nexus between governance and the economic impact of whale-watching. The case of the coastal lagoons in the El Vizcaíno Biosphere Reserve, Baja California, Mexico. Ocean and Coastal Management, 162, 46–59. https://doi.org/10.1016/j.ocecoaman.2018.04.016

In this study, you could get additional evidence for the importance of websites for choosing WW tour operators (see Table 2): 

Lissner, I. & Mayer, M. (2020) Tourists’ willingness to pay for Blue Flag's new ecolabel for sustainable boating: the case of whale-watching in Iceland, Scandinavian Journal of Hospitality and Tourism, 20:4, 352-375, DOI: 10.1080/15022250.2020.1779806

Thank you for suggesting this material. We added information from these papers in the manuscript and the respective references.

Introduction:

[12] Heenehan, H., Basurto, X., Bejder, L., Tyne, J., Higham, J. E., & Johnston, D. W. 2015. Using Ostrom's common-pool resource theory to build toward an integrated ecosystem-based sustainable cetacean tourism system in Hawaii. Journal of Sustainable Tourism, 23(4), 536-556.

[21] Mayer M, Brenner L, Schauss B, Stadler C, Arnegger J, & Job, H. The nexus between governance and the economic impact of whale-watching. The case of the coastal lagoons in the El Vizcaíno Biosphere Reserve, Baja California, Mexico. Ocean & Coastal Management, 2018; 162, 46-59.

[22] Higham J, Bejder L, Allen SJ, Corkeron PJ, & Lusseau D. Managing whale-watching as a non-lethal consumptive activity. Journal of Sustainable Tourism, 2016; 24(1), 73–90. https://doi.org/10.1080/09669582.2015.1062020

Section 4.2

[69] Lissner, I. & Mayer, M. (2020) Tourists’ willingness to pay for Blue Flag's new ecolabel for sustainable boating: the case of whale-watching in Iceland, Scandinavian Journal of Hospitality and Tourism, 20:4, 352-375, DOI: 10.1080/15022250.2020.1779806 

[2] Higgins-Desbiolles F. The “war over tourism”: challenges to sustainable tourism in the tourism academy after COVID-19. Journal of Sustainable Tourism, 2000; 29(4), 551-569.

https://doi.org/10.1080/09669582.2020.1803334 => Surely not published in 2000 but rather in 2020?

[4] Rutty M, Gössling S, Scott D, Hall CM. The global effects and impacts of tourism: an overview. The Routledge handbook of tourism and sustainability, 2005; 54-82. =>Editors? Publisher?

[11] Hoyt E. Whale watching 2001: Worldwide tourism numbers, expenditures, and expanding socioeconomic benefits. 2001 => publisher?

[27] Parsons ECM. The negative impacts of whale-watching. Journal of Marine Biology, 2012. https://doi.org/10.1155/2012/807294 => Article number? Or Volume/Issue/Pages?

[45] Moscardo G, Saltzer R. Understanding tourism wildlife interactions. Sustainable Tourism Cooperative Research, 2005; 36. => A journal or an edited volume?

We carefully checked the reference list and modified the references as pointed out by the reviewer. Other minor improvements were incorporated.

 

Reviewer #2: 

This manuscript aims to understand operators’ perceptions about some mitigation actions to reduce the impacts of the activity on cetaceans, also contrasting the evidence with three experts in the field. This is particularly important for the whale-watching tourism sector since, as the authors underline, there is still scant research concerning operators’ opinions.

Overall, I would like to invite the authors to rethink the manuscript and be more ambitious, i.e., I encourage them to reflect on where to put more focus and emphasis. It might be a reasonable contribution to the field. However, I'll suggest some improvements and ask the authors for a major paper revision.

We thank the reviewer for the encouraging assessment. We have revised the document reflecting on the ambition of the research, and have incorporated all changes as indicated by the reviewer. As a result, we have largely modified the title, abstract, introduction, discussion and conclusions sections. We have emphasized the methodology combining perceptions and the SCI as a novel contribution to the field, as well as the continuous training framework as the main point of novelty to enhance a more sustainable whale-watching practice.

Title

In my opinion, the title is pretty general, although the work has the potential to be summarised in a more attractive and engaged title. I invite the author to consider re-write it.

The title was changed following this suggestion: “The role of operators in sustainable whale-watching tourism: proposing a continuous training framework“.

Abstract

After considering the suggestions, I also invite the authors to re-do it.

The abstract has been modified to reflect the focus on our results and the continuous training framework.

Introduction

As you know, the introduction is one of the most important sections of a manuscript, motivating your research. That is why I encourage the authors to revise the introduction and provide some more order and conciseness (see, e.g., the third paragraph).

The authors should be consistent with the terminology they use around tourism. In the abstract, you talk about “green tourism”. At the same time, you define whale-watching as wildlife-based tourism considered a form of ecotourism since it should adopt the sustainable use of cetaceans (a definition which is definitely right). On the other hand, in the fourth paragraph, you jump into coastal and marine tourism. Please, make it clearer.

Still about the fourth paragraph (lines 75-88). You cite Olszewski-Strzyzowski [31] to talk about some existing good practices for promoting sustainability. However, I think there are ample examples in the whale-watching literature focusing on this issue that you are not citing. Please, reconsider doing a more specific literature review in this regard (overall, along with all the manuscript).

We thank the reviewer for pointing out the terminology issue and for suggesting improvements to the structure of the introduction. Most of the introduction has been rewritten to reflect this and other suggestions from the reviewer, highlighting the research gaps and the novelty of the approach. The suggestions have been followed and a revised introduction includes:

- A revised version of the terminology regarding tourism. We have decided to focus on ecotourism, introducing it at the beginning of the introduction, and we thus have rephrased and deleted mentions to, e.g., “green tourism”.

- The third paragraph has been completely rewritten to reflect the reviewer´s comment; a new, shorter version includes more literature on good practices, and why these are needed in whale-watching context.

Likewise, you begin to talk about online websites straightforwardly. My recommendation is to introduce this point by talking about the importance of how communication impacts consumer expectations and projections about the activity. Please, see the following articles, especially the first one, which is in line with one of your research aims.

- Judge, C., Penry, G. S., Brown, M., & Witteveen, M. (2020). Clear waters: assessing regulation transparency of website advertising in South Africa’s boat-based whale-watching industry.

- Bertella, G. (2019). Close encounters with wild cetaceans: Good practices and online discussions of critical episodes.

- Finkler, W., & Higham, J. E. (2020). Stakeholder perspectives on sustainable whale watching: A science communication approach.

We thank the reviewer for suggesting these papers. We incorporated them in the introduction in the part related to websites and the importance of communication, highlighting one of the main research gaps (fourth paragraph, lines 47-55):

“Communication impacts consumer expectations and projections about the activity. Websites are increasingly used by whale-watching companies as tools to reach a broad, international audience informing tourists and tailoring expectations before the trip [30], [31], [32], [33], [34]. Multiple researchers have sought to analyze content from websites for promoting nature-related tourism [31] [32] [33], but there are no defined standards or globally-accepted methods to evaluate website content [35], [36], [37]. A tool to evaluate web communication in nautical tourism (defined as one of the components of marine tourism) was proposed to include completeness, correctness, and accuracy of website content [35]. There is a need for more studies analyzing if aspects related to sustainable tourism are effectively communicated on whale-watching websites [e.g., 33].”

With regard to the fifth paragraph, please see the following articles, focusing on operators:

- Curtin, S. (2010). Managing the wildlife tourism experience: The importance of tour leaders.

o Despite it is quite old, it should be recommendable to read it, also suitable for increasing the background about operators’ role.

- Hoarau, H., & Kline, C. (2014). Science and industry: Sharing knowledge for innovation.

- Hoarau-Heemstra, H., & Eide, D. (2019). Values and concern: Drivers of innovation in experience-based tourism.

We followed the suggestion and included these studies to the corresponding paragraph about perception (lines 56-69), including:

“Among the stakeholders involved in whale-watching activities, operators play a critical role in the practical activity, and communicating the importance of marine fauna conservation and guiding and inspiring tourists (e.g., [40]; [41]). “

“Whale-watching operators’ high satisfaction in conducting the activity is recognized as critical to achieve sustainability [42], [46], [47], [48].”

In addition, what do you refer with “categories of whale-watching”? I think you are talking about the attributes or aspects that define the experience and/or complement it. In that sense, I invite you to read the following publications about IPA (similar to Tepsich et al.), which would probably help you provide further information and some examples in this regard.

- Bentz, J., Lopes, F., Calado, H., & Dearden, P. (2016). Enhancing satisfaction and sustainable management: Whale watching in the Azores.

- Suárez-Rojas, C., Hernández, M. G., & León, C. J. (2023). Segmented importance-performance analysis in whale-watching: Reconciling ocean coastal tourism with whale preservation.

Thank you for pointing this out. “Whale-watching categories” were defined in Fortuna, C., Canese, S., Giusti, M., Lauriano, G., Mackelworth, P., & Greco, S. (2004). Review of Italian whale-watching: status, problems and prospective. Proceedings of the International Whaling Commission Scientific Committee, ed. IW Commission (Sorrento: Cambridge University Press), 15., and used also in Tepsich P, Borroni A, Zorgno M, Rosso M, Moulins A. Whale Watching in the Pelagos Sanctuary: Status and Quality Assessment. Frontiers in Marine Science, 2020; 7, 596848. https://doi.org/10.3389/fmars.2020.596848. We made it clearer in the introduction, and used the suggested references.

 I also suggest you summarise the next paragraph (lines 98-107) and complement, in that case, the methodology section. Finally, concerning the last three research questions/ aspects, I invite you to emphasise the last one (see my comments in the discussion)

We have modified the last part of the introduction by pointing to the novelty of our approach as well as emphasizing the third research question (lines 71-81):

“This study investigates and evaluates whale-watching operators´ perception of the impacts on marine fauna related to their activities, and proposes a framework for continuous training devoted to whale-watching operators and involving scientific knowledge. A novel compound approach consisting of a combination of questionnaires and data obtained from website analysis is used to tackle the following aspects:

1. what whale-watching operators know about the acoustic impact of their activity on marine mammals, 

2. how whale-watching operators communicate through their websites regarding tourism activity and its impacts on the marine ecosystem,

3. strategies to enhance the sustainability of whale-watching tourism with selected experts in the field.”

Methodology

Section 2.1. Please, I would like you to provide some additional information about the operators. Did you approach them only once? Or did you send them different rounds of emails to remember/ encourage completing the survey? On the other hand, did you pre-test the questionnaire or organise a focus group to validate it? And the last question, is your final sample representative of the target population (Canadian whale-watching operators)?

The questionnaire was defined and discussed within the working team (including social scientists and the ethical department at Stockholm University) and the experts involved in the study. Our sources indicate a total of 91 whale watching companies in Canada. We received 26 responses, which equals to a 29%, assuming that one person per company has responded. However, the survey is anonymous and does not support this consideration. Our results cannot therefore be considered representatives in statistical terms, and a sentence is included in the discussion, first paragraph (lines 303-306):

The responses collected from whale-watching operators are not statistically representative of the entire population of whale-watching operators in Canada, preventing us from conducting a statistical analysis of the results, but nonetheless provide useful insights to understand their point of view (section 4.1).

Section 2.2. Put more emphasis on the Sustainable Communication Index. It is interesting!

More information was added about the SCI. In particular, the importance of the SCI was highlighted by mentioning it in the section title, 4.2: “Web communication and outreach: the Sustainable Communication Index as a way to enhance whale-watching tourism“.

The section was improved with the following sentence that underline the potential of SCI (lines 395-397):

“By defining a Sustainable Communication Index (SCI), we propose a method to estimate the presence of specific information that makes the content complete, accurate, and correct for the whale-watching Canadian context (section 2.2)”. 

An additional discussion was added to section 4.2 to highlight the potential of SCI (lines 405-410):

Additional research is required in order to assess the quality of tourism websites’ content and define specific guidelines on the information to be included: the SCI could be generalized for other areas and/or similar tourism activities, and the website analysis could be extended from text and images to other content sources (e.g., comments, social media profiles). In the presence of a large number of websites, it would benefit from an automated approach (e.g., using content scraping and image processing).

The SCI was also mentioned in the conclusions as it is one of the contributions of this work (lines 484-486): 

“To evaluate website content, we developed a Sustainable Communication Index (SCI) that showed that in some areas, e.g., British Columbia, websites provided more information on marine life and codes of conduct”. 

At some point, in the introduction or the methodology, I’m missing some specific paragraph or section explaining the impacts you include in the questionnaire (short literature review). Suppose you compare maritime transport vessels and whale-watching ones. Do you think the issues you are citing are representative impacts or problems to be prioritised in the context of whale-watching (e.g., the introduction of invasive species)? In this regard, the way you formulated the question (ID 3), in my opinion, could be being misunderstanding. I mean, if you are referring to ships overall, these are really significant impacts; however, are with respect to whale-watching vessels too? Following the example of invasive species, I do not consider whale-watching ships critically contributing to this issue (they do not navigate such long distances and transit through different oceans). That is why I encourage you to explain somewhere the impacts you are studying.

This is indeed a very important point and we thank the referee for the insightful comment. As suggested, we added a brief explanation regarding impacts to section 2.1, keeping in mind that we are not studying potential impacts but how potential impacts are perceived by respondents. In fact, for this study we decided not to select more widespread impacts and let respondents select what they think are the worst ones. 

The newly added text with the explanation reads as follows (lines 113-125):

“Perception of impacts questions (ID 3-5): ID 3 was prepared to evaluate operators' perception and knowledge of vessels' general impacts. Vessels impose a variety of impacts on marine ecosystems. Chemical pollution from vessels is produced mainly by fuel, oil, and anti-fouling treatments discharged from motorized vessels and can reduce water quality or enhance accumulation in sediments. While 4-stroke engines are cleaner, 2-stroke engines, usually used on small boats, produce a higher amount of chemical pollution [51]. Moreover, alien species might be transported on hulls creating a major conservation issue [52]. Ship-strikes can cause whale mortality and have been increasing worldwide due to the increase in ship speed [53]. In addition, shipping noise is of great concern due to a steady increase in traffic. Underwater noise produced by motorized vessels can hinder marine fauna behavior and induce acoustic masking [54]. In order to tackle this issue and find efficient mitigation measures, all the stakeholders involved should be at least aware of the problem. “

Results

Section 3.1. Following my aforementioned comments, why do you think “all of them” was the correct option to be selected? (lines 226-227). 

We think that option was correct because ID 3 was thought of as a question about general vessels impact. The newly added paragraph (from the response above) clarifies this point.

Section 3.2. Could be any possibility to relate the operator’s opinion with the information they provide on their websites? That is, could you confirm if there is any positive relation between those websites that mention aspects about the distance and other sustainable questions with those operators who are more concerned about the whale-watching impacts or the strategies to face them? This would be interesting to know.

We agree that this information would be extremely valuable and we added this point to the discussion (lines 402-410):

“Analyzing operators' websites in combination with their opinions on whale watching impacts and sustainability was not possible with the data presented here. A combined analysis would provide additional information on how they relate and support the identification of positive relations between information provided and positive behaviors”. 

We sent the survey to the same whale-watching companies for which we analyzed the websites, but we did not collect any personal information on the respondents that allow matching their replies with the analyzed websites. Thus, it is not possible to associate respondents with their websites. Results for regions with less than five websites were not shown. Similarly, results of the questionnaire were only shown and discussed for BC and NS, where more responses were collected. This was done to avoid the companies being potentially recognized. A joint analysis of operators' websites and opinions on whale watching impacts and sustainability would be interesting and is currently considered for future work

Discussion

Section 4.1. Lines 324-332. I encourage you to look for some papers in whale-watching literature that have segmented whale-watching consumers, identifying different profiles, from those who effectively expect close encounters to those who are more environmentally friendly.

As advised, a paragraph on segmented consumers have been added at the beginning of section 4.1 (lines 336-338). 

“Some studies (e.g., [59]) identify two types of tourists: specialized whale watchers, who gave the lowest importance to observing whales, and recreationist whale watchers who gave the lowest rate of importance to “whale culture and preservation”. 

Line 337-338. From operators’ responses, are you sure you could affirm that they are overall low-aware? As you later underline from experts’ opinions, it could probably be in relation to (the lack of) information. Please, also consider my previous comments about the impacts.

We agree with this point and we have modified the sentence to reflect the reviewer’s suggestion adding (lines 342-346, addition in bold):

“Impacts produced by biofouling and chemical pollution are perceived similarly by companies operating in BC and NS (Table 4A), showing an overall low awareness of general impacts of vessels on marine fauna-likely due to lack of information-, but an overall high awareness of impacts caused by vessel noise and ship strikes”.

In addition, a sentence mentioning vessels’ general impacts on marine ecosystems was added (as also mentioned in the materials and methods), as advised.

Lines 349-351. It would be interesting if you were able to provide an in-depth argument and relate these regulations with the specific areas of the responding operators (and even operators’ answers). 

Some information regarding regulations extensions have been added (lines 353-361).

“More than half of the analyzed Canadian whale-watching websites mention existing rules regarding keeping a minimum or suggested distance from the cetaceans, which suggests that they are prone to observe them (Table 5). BC operators claim to be more conservative when approaching animals at a distance greater than 100 m compared to NS operators. The Fisheries and Oceans Canada (DFO) regulations are different in case endangered species are present in the area [61]. Vessels should keep a minimum of 100 m from most whales, but stricter regulations can apply in cases of threatened species. For example, a distance of 400 m should be kept when encountering endangered cetaceans in the St. Lawrence estuary and Saguenay river or killer whales in southern BC coastal waters. Thus, some respondents might refer to these extensions of the rules instead of the general ones”.

Section 4.2. 2nd paragraph (lines 380 – 394). Please, check this paragraph. I consider some information quite repetitive.

We reorganized the paragraph reducing the redundancy, shortened the section to reduce repetition and to highlight the novelty presented by the SCI. The paragraph now reads (lines 392-402):

“The overall incomplete awareness of the wide range of impacts of whale watching (section 3.1 and 4.1) is also reflected in the website content analysis. Web communication quality (defined in terms of completeness, correctness, and accuracy of website content) was found to be poor in Italian tourist ports, underlying similar issues [37]. By defining a Sustainable Communication Index (SCI), we propose a method to estimate the presence of specific information that makes the content complete, accurate, and correct for the whale-watching Canadian context (section 2.2). In fact, few websites mentioned whale-watching impacts (Table 5) and minimum distance to be kept when encountering a whale (or an explanation of its importance). A low number of mentions does not necessarily imply a non-virtuous behavior but could mean that the information is considered irrelevant or is too obvious. However, details on marine ecosystems are deemed interesting for tourists by most respondents, Table 4B)”.

On the other hand, please revise lines 395-405. It is quite confusing the way you are presenting the discussion. Some questions look as if they belonged to the previous section.

Following this suggestion, we moved most of the section to the previous and following ones. 

The discussion has been reorganized, moving some sentences to the other sections. The initial part of the discussion now provides an overview of the topics discussed in order to help readers and clarify the message.

Lines 409-413. Have you found any evidence in this or another context that relates the information provided on the websites to consumer satisfaction? Are both forms (website and onboard) of providing information about whale biology, regulations, etc., a precondition for a good whale-watching experience, or complementary? It would be interesting if you could discuss something in this regard. 

The presence of information on ecology and biology is found to be related to consumer satisfaction during trips (e.g., Orams, 2000). However, this has not been demonstrated for websites and the topic is not . As also stated by Law R, Qi S, Buhalis D. Progress in tourism management: A review of website evaluation in tourism research. Tourism management. 2010 Jun 1;31(3):297-313 “As a newly emerging research area, website evaluation has no globally accepted definition yet.” Related information regarding this topic have been highlighted in the Introduction and section 4.3 (e.g., “Whale-watching operators’ high satisfaction in conducting the activity is recognized as critical into achieving sustainability”), as suggested by the reviewer.

In our manuscript, we check the presence of this information and relate it with virtuous (and sustainable) communication. As also stated by Law et al., 2010, several knowledge gaps exist on tourism website evaluation; therefore, the section of this work focusing on websites and communication might serve as a necessary step in bridging this gap, especially due to the lack of recent findings on these topics.

However, future work should assess if the presence of such information on websites can contribute to increasing or managing consumer satisfaction. This might also depend on the different formats in which information is provided (e.g., text, videos, etc.).

Section 4.3. To what method are you referring in the title? My opinion in this section is that you are providing an interesting contribution towards further directions in whale-watching (Fig 7). However, although it has potential, it is not clearly presented, and therefore, its originality is not easily identified. My suggestion is to refocus the paper around this method, being the empirical results evidence that motivates it.

The discussion was reorganized in order to give more emphasis to our contribution in terms of including whale watching impacts in a continuous framework that involves operators (4.3). In this work, we use questionnaires and website analysis to identify what are the knowledge or perception gaps in whale watching operators and the communication strategies adopted for websites (via the sustainable communication index, SCI, also a novel approach). The two methods are simple and subjected to multiple assumptions, but drive us to our final conclusion, i.e. that a continuous training framework is needed for operators and should be based on a) their overall perception and knowledge on the activity, and in particular on the impacts and b) on evidence provided by websites analysis (using the SCI). We worked on better linking our results (operators’ questionnaires, website analysis, and experts’ questionnaires) to the proposed framework and highlighted the novelty of our approach. This is now also reflected in the title that makes direct reference to the conclusion rather than on the methods.

Conclusions.

You do not cite here the contribution of section 4.3, and you could reconsider that. Overall, it would be best if you were more specific about your (theoretical and empirical) contributions and future directions for the sector.

Finally, what are the limitations of this study?

All the best with your revisions!

We have rewritten and redistributed the discussion to reflect this and previous comments about the contributions of our work. The conclusion section has been updated correspondingly, and we have added some reflections about the limitations of the study. In this new version of the manuscript we highlight the use of our combined approach (questionnaires and website analysis) to evaluate the potential integration of a further step in the Orams’ model focusing on whale-watching operators' impact awareness. According to the surveyed experts, increasing outreach activities directed to whale-watching operators could help improve their awareness regarding impact and motivate operators to reduce them.

 

Additional responses related to journal requirements

Journal Requirements:

2. You indicated that ethical approval was not necessary for your study. We understand that the framework for ethical oversight requirements for studies of this type may differ depending on the setting and we would appreciate some further clarification regarding your research. Could you please provide further details on why your study is exempt from the need for approval and confirmation from your institutional review board or research ethics committee (e.g., in the form of a letter or email correspondence) that ethics review was not necessary for this study? Please include a copy of the correspondence as an ""Other"" file.

The research did not involve any sensitive personal data, as defined in the EU General Data Protection Regulation (GDPR). Therefore, the research does not require ethical review according to the Swedish Ethical Review Act. The research protocol and questionnaires were prepared based on input from the Ethics support function at Stockholm University, in order to ensure no sensitive data would be processed. The Ethics support function confirms that there is no need for ethical review in an email, see below (full conversation attached as “Other”):

Research participants were given written information about the project in an information sheet based on the template provided by Stockholm University. Written consent was collected from participants, also based on templates from Stockholm University. No minors were involved in the study.

4. Please include a complete copy of PLOS’ questionnaire on inclusivity in global research in your revised manuscript. Our policy for research in this area aims to improve transparency in the reporting of research performed outside of researchers’ own country or community. The policy applies to researchers who have travelled to a different country to conduct research, research with Indigenous populations or their lands, and research on cultural artefacts. The questionnaire can also be requested at the journal’s discretion for any other submissions, even if these conditions are not met. Please find more information on the policy and a link to download a blank copy of the questionnaire here: https://journals.plos.org/plosone/s/best-practices-in-research-reporting. Please upload a completed version of your questionnaire as Supporting Information when you resubmit your manuscript.

A completed PLOS’ questionnaire on inclusivity in global research is included as Supporting Information in the revised manuscript.

AA was funded by OGS Project BlueSkills: Blue Jobs and Responsible Growth in the Mediterranean, labeled by the UfM and funded by the Italian Ministry of University and Research (MUR) 

AS was funded by the Department of Physical Geography, Stockholm University. Stockholm University supports open access publishing by covering article-processing charges.

The financial disclosure has been inserted in the rebuttal/cover letter.

Funding for this study has been provided by OGS Project BlueSkills: Blue Jobs and Responsible Growth in the Mediterranean, labeled by the UfM and funded by the Italian Ministry of University and Research (MUR) and by the Department of Physical Geography, Stockholm University. However, funding information should not appear in the Acknowledgments section or other areas of your manuscript. We will only publish funding information present in the Funding Statement section of the online submission form. Please remove any funding-related text from the manuscript and let us know how you would like to update your Funding Statement. Currently, your Funding Statement reads as follows: 

AA was funded by OGS Project BlueSkills: Blue Jobs and Responsible Growth in the Mediterranean, labeled by the UfM and funded by the Italian Ministry of University and Research (MUR) 

The amended statement has been added to the cover letter.

7. Your ethics statement should only appear in the Methods section of your manuscript. If your ethics statement is written in any section besides the Methods, please move it to the Methods section and delete it from any other section. Please ensure that your ethics statement is included in your manuscript, as the ethics statement entered into the online submission form will not be published alongside your manuscript. 

The ethics statement has been moved to the Methods section.

8. We note that Figure 1 in your submission contain map images which may be copyrighted. All PLOS content is published under the Creative Commons Attribution License (CC BY 4.0), which means that the manuscript, images, and Supporting Information files will be freely available online, and any third party is permitted to access, download, copy, distribute, and use these materials in any way, even commercially, with proper attribution. For these reasons, we cannot publish previously copyrighted maps or satellite images created using proprietary data, such as Google software (Google Maps, Street View, and Earth). For more information, see our copyright guidelines: http://journals.plos.org/plosone/s/licenses-and-copyright.

To address this comment, we have added the following to the caption which makes direct reference to the data license for OpenStreetMap data following their guidelines: " Map data from OpenStreetMap available from https://www.openstreetmap.org (Openstreetmap contributors, 2023) under the Open Data Commons Open Database License (ODbL)". We have added the reference “OpenStreetMap contributors (2023), Planet dump retrieved from https://planet.openstreetmap.org on 1/12/2022. Data licensed under the Open Data Commons Open Database License (ODbL) by the OpenStreetMap Foundation (OSMF).".

We have added captions for our Supporting Information at the end of the manuscript.

---

## [Decision Letter · Decision Letter 1]

16 Oct 2023

PONE-D-22-32212R1The role of operators in sustainable whale-watching tourism: proposing a continuous training frameworkPLOS ONE

Dear Dr. Scaini,

Thank you for submitting your manuscript to PLOS ONE. After careful consideration, we feel that it has merit but does not fully meet PLOS ONE’s publication criteria as it currently stands. Therefore, we invite you to submit a revised version of the manuscript that addresses the points raised during the review process.

We look forward to receiving your revised manuscript.

Kind regards,

Vitor Hugo Rodrigues Paiva, Ph.D.

Academic Editor

PLOS ONE

Journal Requirements:

Reviewers' comments:

Reviewer's Responses to Questions

**Comments to the Author**

1. If the authors have adequately addressed your comments raised in a previous round of review and you feel that this manuscript is now acceptable for publication, you may indicate that here to bypass the “Comments to the Author” section, enter your conflict of interest statement in the “Confidential to Editor” section, and submit your "Accept" recommendation.

Reviewer #1: All comments have been addressed

Reviewer #3: (No Response)

2. Is the manuscript technically sound, and do the data support the conclusions?

Reviewer #1: Yes

Reviewer #3: Yes

3. Has the statistical analysis been performed appropriately and rigorously? 

Reviewer #1: Yes

Reviewer #3: N/A

4. Have the authors made all data underlying the findings in their manuscript fully available?

Reviewer #1: Yes

Reviewer #3: Yes

5. Is the manuscript presented in an intelligible fashion and written in standard English?

Reviewer #1: Yes

Reviewer #3: Yes

6. Review Comments to the Author

Reviewer #1: Thank you very much for your revised version of your manuscript and your detailed replies to my comments and criticism!

“The research was set up during the COVID-19 pandemic lockdown. Therefore, online questionnaires were the only possible option.” => Really? What about Zoom interviews or phone calls? I personally know a lot of researchers using these approaches during the lockdowns… Anyway…

In Table 4 it must “Tourists’ preferences…”, so with a genitive form, thanks.

In Reference 4 Line 540 there is a typo: “G$ssling” instead of “Gössling”, please correct.

Reference 21: doi is missing.

All in all, I think the manuscript has been much improved and could be accepted now for publication.

Reviewer #3: The manuscript provides the perspective of Canadian whale-watching tour operators on responsible behaviour during the tours and their awareness of the potential impacts that the activity may cause. This view is complemented by the analysis of operators’ website contents and contrasted with the opinion of three experts. The study highlights the necessity of considering operators' perceptions to identify weaknesses and suggests a continuous training framework to address them. Considering that whale-watching is continuously increasing worldwide, and it has a large socio-economic impact in several countries, it is essential to find ways to understand how their practices can be responsibly improved and their potential impacts mitigated.

Overall the manuscript is well written and structured, with notable improvements after the first revision. The methodology is based mostly on online surveys, and one of the main constraints is the limited number of surveys considered (as previous reviewers have pointed out, and the authors have already argued in the text). However, the novelty of the study relies on the combination of approaches to address the same topic, and I particularly appreciate the creation of a simple index to objectively quantify the online communication strategies of the companies.

I suggest some minor revisions before publication to clarify some details in the text.

ABSTRACT

P19, L13. Following the style of the (original) title (positive!), I suggest rewriting this sentence to start with something positive too (e.g. "Whale watching is considered a form of green tourism"... "the threats you say etc demand a better understanding to develop a more sustainable industry").

INTRODUCTION

P20, L30. To start in a more assertive way (instead of having the “notwithstanding” among the first words). E.g. “Accompanying the rise of ecotourism”..

P20, L34. “…has been increasing worldwide…” since when? Add some temporal context.

P20, L53. improve -> improving

P21, L56. Delete “and projections”

METHODS

The advantages you cite in the response to Reviewer #1 (P96) are of interest to support your methodology choice. I would add them to the manuscript.

P27, L195. I only realised when reading the results what these columns refer to. I don't think the column indication is needed here. Clear enough (even clearer!) without it.

P28, L215. The section you refer to should be section 2.1 (instead of 2.2.1).

RESULTS

P23, L119. Regarding ID 1, it should be good to briefly indicate why you used these distances, I would say based on current guidelines/legislation? Also check your reply to Reviewer#1 (P99 – Table 1, ID 1)

P23, . Regarding ID 2, I think it would be also of interest to include the info of your response to Reviewer #1, P99 ID 2 about the increasing scale, as it helps to design further studies avoiding the same problem.

P29, L226-228. Do "companies", "operators" and "respondents" mean the same? it sounds a bit confusing here. I wonder if a company may have different operators, or if several respondents may come from the same company/operator? I suggest briefly clarifying this, or standardising the names for clarity.

P29, FIG.1. Delete the names of the provinces where you don't have data to avoid noise in the image. In the caption, you should state the full name of the ones you use, i.e. BC, QC, NB, NS and NL (as you already did with BC and NS).

P29, L239. “(The..)” -> (the..)

P30, L250. Is the same ONE company in both provinces? or one company in each of them? Please clarify.

P30, L252. “Most perceived issues have similar percentage…” In both areas?

P30, L262. “In NS waters, operators stated that they decrease the vessel’s speed (n=5) and implement mandatory avoidance of feeding and breeding areas”. But do they already comply with this? Or these are the options they prefer? Please, clarify.

P31, L276. Delete (section 2.2). And if you already have added the name on the Fig. 1 caption, here you keep only the QC (as for BC). The same with the next 2 lines.

P32, L284. As it refers to the same question, keep it in one sentence: “…general; in NS…”, and put the reference to the table for this, which should be Fig. S3A, instead of Fig S3.

P32, L285 and L289. Accordingly, Fig. S3B and Fig. S3C (at the end of the paragraph).

P32, L287. As it refers to the same question, keep it in one sentence: “…too close and 53%...”

P32, L293. Results from “operators” questionnaire...

P33, L307. I would state here the highlights of this answer. And keep the S4 as it is. Otherwise, this topic loses impact in the manuscript, as no info is provided about these results.

DISCUSSION

P33, L315. As you are now in the discussion, I suggest re-writing differently this sentence (as it is now, it sounds more like methods). As an example: "A Sustainable Communication Index is introduced, as a novel tool to objectively analyse the online whale watching communication strategies (section 4.2).”

P33, L317. “The questionnaire for whale watching operators can…”

P34, L340. “…Orams stated that 35% of whale-watching tourists…” From where? Worldwide?

P35, L369. Indicate the province between brackets for the St. Lawrence estuary and Saguenay River.

P36, L380. Be Whale Wise (big letter). Can you explain briefly what this is, or what’s the main goal or achievement of these guidelines?

P36, L381. Please confirm the Table number, I would say that you want to refer to Table 5 instead of 2.

P36, L398-399. Incomplete sentence.

P37, L412. “Analyzing […] was not possible…” Briefly explain why

P37, L423-424. Delete or re-write “and the importance of managing passenger expectations [46]”.

P38, L436-441. I suggest explaining the Orams model before you propose modifications (i.e. L436).

SUPPORTING INFORMATION

S5 and S6 are cited in the text before other S figures. Please consider re-numbering for consistency and organization.

P2, L16. Whale-watching operators questionnaire. (We already assume that a questionnaire has questions and that the table should be below).

P4, L21. “Each question…” This sentence is not correct. Fix or maybe simplify like: Questions between both questionnaires are linked by their identifier (ID).

P6, L26-32. As you have some common aspects for all the sections of the figure, please consider writing these first and indicate where these results come from: “Main results from the whale-watching companies website analysis. British Columbia (BC) on the left column, Nova Scotia (NS) in the middle and the whole dataset for Canada on the right. (A) Mention of impacts on marine fauna. Results for Nova Scotia (NS), Prince Edward Island (PE), and Nunavut (NU) are not displayed. (B) Mention of best practices. Results for Prince Edward Island (PE) and Nunavut (NU) are not displayed. (C) Mention of distance kept from mammals. Results for Manitoba (MB) and Quebec (QC) are not displayed. For the whole Figure: NL = Newfoundland and Labrador, BC = British Columbia, MB = Manitoba, NB = New Brunswick, QC = Quebec, MB = Manitoba and NS = Nova Scotia.”

7. PLOS authors have the option to publish the peer review history of their article (what does this mean?). If published, this will include your full peer review and any attached files.

Reviewer #1: No

Reviewer #3: **Yes: **Laura González García

---

## [Author Response · Author response to Decision Letter 1]

7 Dec 2023

Academic Editor

PLOS ONE

Journal Requirements:

→ The reference list was reviewed and no reference was removed. Some minor corrections were undertaken thanks to the reviewer´s work and a careful check for typos.

Reviewer #1

Thank you very much for your revised version of your manuscript and your detailed replies to my comments and criticism!

→ Thank you for providing such great and constructive comments.

“The research was set up during the COVID-19 pandemic lockdown. Therefore, online questionnaires were the only possible option.” => Really? What about Zoom interviews or phone calls? I personally know a lot of researchers using these approaches during the lockdowns… Anyway…

→ Yes, we agree that questionnaires were not the only possible option, we could have presented this point more carefully. This response has been mentioned by Reviewer 3 as well. We have added some more details regarding the choice of the methodology in the main text, and amended this point.

In Table 4 it must “Tourists’ preferences…”, so with a genitive form, thanks.

→ We have incorporated this correction.

In Reference 4 Line 540 there is a typo: “G$ssling” instead of “Gössling”, please correct.

→ Thanks for checking the references, we have amended the typo.

Reference 21: doi is missing.

→ We have added the doi.

All in all, I think the manuscript has been much improved and could be accepted now for publication.

Reviewer #3 

The manuscript provides the perspective of Canadian whale-watching tour operators on responsible behaviour during the tours and their awareness of the potential impacts that the activity may cause. This view is complemented by the analysis of operators’ website contents and contrasted with the opinion of three experts. The study highlights the necessity of considering operators' perceptions to identify weaknesses and suggests a continuous training framework to address them. Considering that whale-watching is continuously increasing worldwide, and it has a large socio-economic impact in several countries, it is essential to find ways to understand how their practices can be responsibly improved and their potential impacts mitigated.

Overall the manuscript is well written and structured, with notable improvements after the first revision. The methodology is based mostly on online surveys, and one of the main constraints is the limited number of surveys considered (as previous reviewers have pointed out, and the authors have already argued in the text). However, the novelty of the study relies on the combination of approaches to address the same topic, and I particularly appreciate the creation of a simple index to objectively quantify the online communication strategies of the companies.

I suggest some minor revisions before publication to clarify some details in the text.

→ We thank the reviewer for their work. Below we address each point and clarify the details as suggested. In addition, we have carefully revised the whole manuscript to improve the text. 

ABSTRACT

P19, L13. Following the style of the (original) title (positive!), I suggest rewriting this sentence to start with something positive too (e.g. "Whale watching is considered a form of green tourism"... "the threats you say etc demand a better understanding to develop a more sustainable industry").

→ We have modified the sentence to give a more positive tone as suggested. 

INTRODUCTION

P20, L30. To start in a more assertive way (instead of having the “notwithstanding” among the first words). E.g. “Accompanying the rise of ecotourism”..

→ We have modified the sentence accordingly. It now reads (lines 29-31):

“Tourist activities impact ecosystems directly, e.g., through habitat degradation, pollution, and loss of biodiversity, and indirectly by affecting ecosystem services provision, and despite the rise of ecotourism practices (refs)”.

P20, L34. “…has been increasing worldwide…” since when? Add some temporal context.

→ We have added “since the early ´90s” (as reported by the referenced literature).

P20, L53. improve -> improving

P21, L56. Delete “and projections”

→ We have modified the sentences at page 20 and 21 as suggested.

METHODS

The advantages you cite in the response to Reviewer #1 (P96) are of interest to support your methodology choice. I would add them to the manuscript.

→ We have improved the information to support the methodological choice in the main text as suggested (lines 331-332):

“Online questionnaires were chosen because of their time and cost efficiency, flexibility, lowered interviewer bias, and due to the unrestricted geographic coverage (Chang & Vowles, 2013)”.

Chang, T. Z. D., & Vowles, N. 2013. Strategies for improving data reliability for online surveys: A case study. International Journal of Electronic Commerce Studies, 4(1), 121-130.

P27, L195. I only realised when reading the results what these columns refer to. I don't think the column indication is needed here. Clear enough (even clearer!) without it.

→ The reference to the columns was deleted as suggested.

P28, L215. The section you refer to should be section 2.1 (instead of 2.2.1).

→ The section was corrected.

RESULTS

P23, L119. Regarding ID 1, it should be good to briefly indicate why you used these distances, I would say based on current guidelines/legislation? Also check your reply to Reviewer#1 (P99 – Table 1, ID 1)

→ The following text was added to clarify the choices (lines 138-143): 

“We selected 100 and 200 m based on current guidelines and legislation. Canada's Marine Mammal Regulations mention “keeping a minimum of 100 meters away from most whales, dolphins, and porpoises, and keeping a minimum of 200 meters away if they are in resting position or with their calf.” Other reasonable distances in the upper and lower bound of the prescribed distance (displaying them in the question in an increasing order) were included in order to assess operators' knowledge, and to partially prevent biased responses”. 

P23, . Regarding ID 2, I think it would be also of interest to include the info of your response to Reviewer #1, P99 ID 2 about the increasing scale, as it helps to design further studies avoiding the same problem.

→ A sentence was added to clarify this further (lines 144-145):

“We did not choose an increasing scale to try to avoid bias and discourage respondents towards selecting an intermediate number”.

P29, L226-228. Do "companies", "operators" and "respondents" mean the same? it sounds a bit confusing here. I wonder if a company may have different operators, or if several respondents may come from the same company/operator? I suggest briefly clarifying this, or standardising the names for clarity.

→ the text has been carefully reviewed to improve the wording by using the word “operators” instead of a mix of the three mentioned above. 

P29, FIG.1. Delete the names of the provinces where you don't have data to avoid noise in the image. In the caption, you should state the full name of the ones you use, i.e. BC, QC, NB, NS and NL (as you already did with BC and NS). 

→ Figure 1 was edited to delete the names of the provinces without data. The caption was also modified as suggested. 

P29, L239. “(The..)” -> (the..)

P30, L250. Is the same ONE company in both provinces? or one company in each of them? Please clarify.

→ The sentence was modified to (line 255): 

 “Only one operator in BC and one in NS chose the correct option, “All of them””.

P30, L252. “Most perceived issues have similar percentage…” In both areas?

→ Yes, the sentence was modified to (lines 256-257): 

 “Most of the other issues had a similar percentage of responses in BC and NS, except for chemical pollution, which was indicated 8 times in BC and 1 time for NS.”

P30, L262. “In NS waters, operators stated that they decrease the vessel’s speed (n=5) and implement mandatory avoidance of feeding and breeding areas”. But do they already comply with this? Or these are the options they prefer? Please, clarify.

→ They state that they do so in the questionnaire. The sentence was modified as allows (lines 267-269):

“In NS waters, operators indicated in their responses that they decrease the vessel’s speed (n=5) and implement mandatory avoidance of feeding and breeding areas during the most important times of day for these activities (n=2)”.

P31, L276. Delete (section 2.2). And if you already have added the name on the Fig. 1 caption, here you keep only the QC (as for BC). The same with the next 2 lines.

→This is the first mention to Québec, so we kept the name. In the following lines we also have left the full names of the provinces, a+s this is the first time we mention them in the main text.

P32, L284. As it refers to the same question, keep it in one sentence: “…general; in NS…”, and put the reference to the table for this, which should be Fig. S3A, instead of Fig S3.

→ The sentence has been modified as suggested.

P32, L285 and L289. Accordingly, Fig. S3B and Fig. S3C (at the end of the paragraph).

→ The sentence has been modified as suggested.

P32, L287. As it refers to the same question, keep it in one sentence: “…too close and 53%...”

→ The sentence has been modified as suggested.

P32, L293. Results from “operators” questionnaire…

→ The sentence has been modified as suggested.

P33, L307. I would state here the highlights of this answer. And keep the S4 as it is. Otherwise, this topic loses impact in the manuscript, as no info is provided about these results.

→ To address this point, we have revised the paragraph and added a summary of the most important points raised by the experts. The sentence now reads (lines 308-311): 

“The experts chose what they believed to be the most critical aspects for tourists during whale-watching trips among the provided options, and added additional aspects including “enjoying the camaraderie of watching wildlife in a good group”, “getting out on the water”, and “feeling like they experienced something special” (Table S4). “

DISCUSSION

P33, L315. As you are now in the discussion, I suggest re-writing differently this sentence (as it is now, it sounds more like methods). As an example: "A Sustainable Communication Index is introduced, as a novel tool to objectively analyse the online whale watching communication strategies (section 4.2).”

→ The sentence has been modified as suggested.

P33, L317. “The questionnaire for whale watching operators can…”

→ The sentence was amended as suggestd.

P34, L340. “…Orams stated that 35% of whale-watching tourists…” From where? Worldwide?

→ The sentence has been modified to include the spatial scale of the study (lines 344-345):

“In a study conducted in Moreton Island, Australia, 35% of whale-watching tourists were satisfied with the trip even without encountering whales [45]”.

P35, L369. Indicate the province between brackets for the St. Lawrence estuary and Saguenay River.

→ the province of Québec was indicated as suggested.

P36, L380. Be Whale Wise (big letter). Can you explain briefly what this is, or what’s the main goal or achievement of these guidelines?

→ We have capitalized the letters and added some more information about the Be Whale Wise guidelines (lines 383-386):

“The Be Whale Wise goal is to educate about codes of conduct and best practices around the whale-watching experience; such best practices were mentioned on many operators´ websites (with sections called “code of conduct”, Table 5)”.

→ We have modified the whole paragraph to explain more clearly in what way whale-watching practices could be more aware of whale´s disturbance.

P36, L381. Please confirm the Table number, I would say that you want to refer to Table 5 instead of 2.

→ We indeed refer to Table 5 and have amended it in the text.

P36, L398-399. Incomplete sentence.

→ The sentence has now been amended; the end of the paragraph now reads (lines 401-404):

“However, younger tourists (for instance in [69] the average age was 38 years) might be more comfortable looking for information online. Not all tourists looking for a whale-watching tour will look for options online, and small companies might advertise their activities only on the docks”.

P37, L412. “Analyzing […] was not possible…” Briefly explain why

→ To explain this point, we have added to the sentence “because anonymity was kept for all participants” (lines 416-417).

P37, L423-424. Delete or re-write “and the importance of managing passenger expectations [46]”.

→ The last part of the sentence was deleted as suggested.

P38, L436-441. I suggest explaining the Orams model before you propose modifications (i.e. L436).

→ This part was modified following this suggestion. The text presenting the modified Orams model was sharpened (lines 441-450):

“The framework of the Orams model [45], initially developed to model tourists’ education, was here adjusted for whale-watching operators’ education, participatory and active training. We modified the Orams model by introducing a step in the training program denominated “Impacts awareness” (blue box, Fig. 2) as part of “The Affective Domain” and “Curiosity.”

SUPPORTING INFORMATION

S5 and S6 are cited in the text before other S figures. Please consider re-numbering for consistency and organization.

→ Following the reviewer’s suggestion, the supporting texts, figures, and tables have been renumbered following the order of appearance in the manuscript.

P2, L16. Whale-watching operators questionnaire. (We already assume that a questionnaire has questions and that the table should be below).

→ Thank you for pointing this out. We re-worded accordingly. Now the caption reads: “S1 Table: Questionnaire for whale-watching operators used in this paper”

P4, L21. “Each question…” This sentence is not correct. Fix or maybe simplify like: Questions between both questionnaires are linked by their identifier (ID).

→ We followed the reviewer’s suggestion and rewrote the sentence as “Questions between both questionnaires are linked by their identifier (ID).”

P6, L26-32. As you have some common aspects for all the sections of the figure, please consider writing these first and indicate where these results come from: “Main results from the whale-watching companies website analysis. British Columbia (BC) on the left column, Nova Scotia (NS) in the middle and the whole dataset for Canada on the right. (A) Mention of impacts on marine fauna. Results for Nova Scotia (NS), Prince Edward Island (PE), and Nunavut (NU) are not displayed. (B) Mention of best practices. Results for Prince Edward Island (PE) and Nunavut (NU) are not displayed. (C) Mention of distance kept from mammals. Results for Manitoba (MB) and Quebec (QC) are not displayed. For the whole Figure: NL = Newfoundland and Labrador, BC = British Columbia, MB = Manitoba, NB = New Brunswick, QC = Quebec, MB = Manitoba and NS = Nova Scotia.”

→ Thank you for suggesting this change. The new caption reads: 

“Main results from the whale-watching companies’ website analysis. British Columbia (BC) is shown on the left column, Nova Scotia (NS) in the middle, and the whole dataset for Canada on the right. (A) Mention of impacts on marine fauna. In the “Canada” column, results for Nova Scotia (NS), Prince Edward Island (PE), and Nunavut (NU) are not displayed. (B) Mention of best practices. In the “Canada” column, results for Prince Edward Island (PE) and Nunavut (NU) are not displayed. (C) Mention of distance kept from mammals. In the “Canada” column, results for Manitoba (MB) and Quebec (QC) are not displayed. For the whole Figure: NL = Newfoundland and Labrador, BC = British Columbia, MB = Manitoba, NB = New Brunswick, QC = Quebec, MB = Manitoba and NS = Nova Scotia”.

---

## [Editor Report · Decision Letter 2]

10 Dec 2023

The role of operators in sustainable whale-watching tourism: proposing a continuous training framework

PONE-D-22-32212R2

Dear Dr. Scaini,

We’re pleased to inform you that your manuscript has been judged scientifically suitable for publication and will be formally accepted for publication once it meets all outstanding technical requirements.

Kind regards,

Vitor Hugo Rodrigues Paiva, Ph.D.

Academic Editor

PLOS ONE
---

## [Editor Report · Acceptance letter]

20 Dec 2023

PONE-D-22-32212R2 

PLOS ONE

Dear Dr. Scaini, 

I'm pleased to inform you that your manuscript has been deemed suitable for publication in PLOS ONE. Congratulations! Your manuscript is now being handed over to our production team.

Kind regards, 

on behalf of

Dr. Vitor Hugo Rodrigues Paiva 

Academic Editor

PLOS ONE